# Beyond Task-Specific Classifiers: In-Context Inference for Time Series Classification Foundation Models

**Juntao Fang** [* 1] **Shifeng Xie** [* 2 3] **Shengbin Nie** [1] **Yuhui Ling** [1] **Yuming Liu** [1] **Zijian Li** [4] **Keli Zhang** [2] **Lujia Pan** [5] **Themis Palpanas** [3] **Ruichu Cai** [1]

## Abstract

Time series classification foundation models are commonly evaluated by freezing a pretrained backbone and fitting task-specific classifiers on extracted representations. While effective, this pipeline entangles representation quality with classifier choice, hyperparameter tuning, and per-dataset optimization, making it less suitable when deployment-time fitting is undesirable. We propose TIC-FM, a deployment-time training-free in-context inference framework that uses labeled support examples as context and predicts query labels without fitting task-specific classifiers. TIC-FM maps time series and labels into an ICL-compatible token space, consolidates long contexts with latent memory, and performs leakage-safe parallel inference via support-only label injection and split-masked reasoning. On the UCR archive, TIC-FM improves average accuracy over representative classifier-fitting baselines, especially in low-label regimes, suggesting that support-conditioned in-context inference is a practical alternative for time series classification.

## 1. Introduction

Time series classification is central to applications such as human movement analysis, clinical monitoring, digital health, and energy systems, where accurate recognition of temporal patterns supports downstream decision-making (Ismail Fawaz et al., 2019). Recent progress in time series foundation models (TSFMs) has made it increasingly feasible to reuse a single pretrained backbone across domains

(Liang et al., 2024). At the same time, in privacy-sensitive or labor-intensive settings such as healthcare, collecting and annotating labeled time series at scale is often costly or infeasible, increasing the need for methods that can adapt to new classification tasks with limited supervision and minimal task-specific optimization (Gao et al., 2025). For classification with TSFMs, this motivates deployment pipelines that can effectively exploit a small labeled support set without fitting a new classifier for every downstream dataset.

Despite this motivation, the prevailing evaluation practice for time series classification foundation models remains the frozen-backbone classifier-fitting pipeline: a pretrained backbone is frozen, representations are extracted, and a task-specific classifier such as an SVM, random forest, or MLP is fitted for each downstream dataset. Although effective, this pipeline entangles representation quality with classifier choice, hyperparameter settings, and per-dataset optimization. As shown in Table 1, the same backbone can yield substantially different conclusions depending on the downstream classifier, complicating both evaluation and deployment when fitting a new classifier is undesirable.

This limitation is especially important in low-resource scenarios. Parametric classifiers can be unstable or prone to overfitting with few labels, while non-parametric rules such as nearest neighbors or centroids impose restrictive distance-based biases. In on-device, latency-sensitive, or privacy-constrained applications, repeated classifier fitting may also be impractical. This raises the question: **can a time series classification model condition directly on labeled support examples and predict query labels without deployment-time parameter fitting?**

We address this question through deployment-time training-free in-context inference. This setting removes deployment-time parameter updates by using support examples for in-context conditioning rather than for fitting a new task-specific classifier. Once pretrained, the model can handle a new classification task through forward computation alone. As illustrated in Figure 1, we replaces per-dataset classifier fitting with support-conditioned in-context inference. We instantiate this paradigm with the **T**ime Series **I**n-Context **C**lassification **F**oundation **M**odel (**TIC-FM**), which maps

---

[1]Guangdong University of Technology, Guangzhou, China [2]Huawei Noah's Ark Lab, Paris, France [3]Université Paris Cité, LIPADE, Paris, France [4]Mohamed bin Zayed University of Artificial Intelligence, Abu Dhabi, United Arab Emirates [5]Huawei Noah's Ark Lab, Shenzhen, China. Correspondence to: Ruichu Cai <cairuichu@gmail.com>.

*Proceedings of the $2^{nd}$ ICML Workshop on Foundation Models for Structured Data*, Seoul, South Korea. 2026. Copyright 2026 by the author(s).

time series into instance-level representations, projects them into an ICL-compatible token space, and reasons over labeled support tokens and unlabeled query tokens. On the UCR archive, TIC-FM achieves higher average accuracy than representative frozen-backbone classifier-fitting baselines, especially in low-label regimes, without fitting per-dataset classifiers.

**Contributions.** Our contributions are threefold: (i) we revisit the frozen-backbone classifier-fitting pipeline for TSFM-based classification and formulate deployment-time training-free in-context inference as an alternative paradigm; (ii) we propose TIC-FM, which combines support-only label injection, latent context consolidation, and split-masked support-query reasoning for leakage-safe parallel inference; and (iii) we validate TIC-FM on the UCR archive through comparisons with representative classifier-fitting baselines, low-label evaluation, context-scaling experiments, efficiency analysis, and ablation studies.

## 2. Preliminary

**Frozen-backbone classifier-fitting pipeline.** A time series classification task provides labeled examples $\mathcal{D} = \{(x_i, y_i)\}_{i=1}^n$, with $x_i \in \mathbb{R}^{l \times T}$ ($l = 1$ for univariate series) and $y_i \in \{1, \ldots, K\}$. In the standard TSFM pipeline, a frozen encoder $F_\psi$ maps each instance to $z_i = F_\psi(x_i) \in \mathbb{R}^q$, and a task-specific classifier $h_\tau : \mathbb{R}^q \to \mathbb{R}^{K_\tau}$ is fitted on the embedded training split. Query prediction is then

$$\hat{y}(x) = \arg\max_{k \in \{1, \ldots, K_\tau\}} [h_\tau(F_\psi(x))]_k.$$

**In-context classification.** We instead treat the labeled training split as a context set $\mathcal{D}_{\mathrm{tr}}$ and the test split as a query set $\mathcal{D}_{\mathrm{te}}$. A deployment-time training-free classifier estimates $p(y_{\mathrm{te}} \mid x_{\mathrm{te}}, \mathcal{D}_{\mathrm{tr}})$ using a fixed model and no target-task parameter updates. This setting keeps task supervision through support labels, but replaces classifier fitting with in-context conditioning. It is therefore a few-label adaptation setting: labels are used as prompts rather than as optimization data for a new task-specific classifier. We realize this goal via in-context inference, as detailed in Section 3.

## 3. TIC-FM Methodology

**Overview.** Following the deployment-time training-free ICL setting in Section 2, TIC-FM predicts query labels by conditioning on labeled support examples rather than fitting a task-specific classifier. Given a support set $\mathcal{D}_{\mathrm{tr}} = \{(x_i, y_i)\}_{i=1}^{N_{\mathrm{tr}}}$ and a query set $\mathcal{D}_{\mathrm{te}} = \{x_j\}_{j=1}^{N_{\mathrm{te}}}$, TIC-FM jointly processes the concatenated support and query instances and produces predictions for all queries in parallel. As shown in Figure 2, the model consists of three modules: a time-series encoder $F_\psi$, a projection adapter $g_\phi$, and an in-context classifier $G_\theta$.

**Time-series feature encoder.** The encoder $F_\psi$ maps each input series $x \in \mathbb{R}^T$ to an instance embedding $z = F_\psi(x) \in \mathbb{R}^q$. Following Mantis (Feofanov et al., 2025), it first constructs patch embeddings $U \in \mathbb{R}^{P \times q}$ by combining two complementary sources of information: local temporal dynamics, extracted by convolutional features from both the raw signal $x$ and its first-order difference $\Delta x$, and patch-level statistics, such as the mean and standard deviation of each patch. These patch tokens provide local evidence for context-based matching, which is particularly useful when labeled support examples are limited. We then apply a ViT-style Transformer (Dosovitskiy, 2020; Feofanov et al., 2025) with positional encodings and a learnable classification token to aggregate information across patches. The output of the classification token is taken as the final embedding $z$, yielding a compact and discriminative representation for downstream in-context classification.

**Projection Adapter.** The encoder embedding space $\mathbb{R}^q$ is not necessarily aligned with the token space expected by the in-context classifier. Since in-context inference is driven by attention weights and token-wise interactions, mismatched feature statistics can degrade conditioning on labeled context. We therefore learn a lightweight projection adapter $g_\phi$ to map instance embeddings into an ICL-compatible representation space. We implement $g_\phi$ as an MLP with Layer Normalization (Ba et al., 2016):

$$g_\phi(z) = W_2 \, \sigma\big(W_1 \, \mathrm{LN}(z)\big), W_1 \in \mathbb{R}^{d_h \times q}, W_2 \in \mathbb{R}^{d \times d_h},$$

where $\sigma$ is GELU (Hendrycks, 2016). The adapter is applied independently to each instance embedding, enabling efficient batch processing of dataset sequences. In our implementation, $d$ matches the internal model dimension of the ICL Transformer.

**In-context classifier.** Given projected support and query embeddings $H_{\mathrm{tr}} \in \mathbb{R}^{N_{\mathrm{tr}} \times d}$ and $H_{\mathrm{te}} \in \mathbb{R}^{N_{\mathrm{te}} \times d}$, the in-context classifier $G_\theta$ operates on the concatenated sequence $H^{(0)} = [H_{\mathrm{tr}}; H_{\mathrm{te}}] \in \mathbb{R}^{N \times d}$, where $N = N_{\mathrm{tr}} + N_{\mathrm{te}}$. Its computation consists of three steps: latent context consolidation, support-only label injection, and split-masked support-query reasoning.

First, to handle long support contexts, we use a Perceiver-style latent memory module with learnable latent tokens $L \in \mathbb{R}^{M \times d}$, where $M \ll N_{\mathrm{tr}}$. The module writes information from support features into the latents and then reads the latent summary back to refine all tokens:

$$\tilde{L} = \mathrm{Write}(L, H_{\mathrm{tr}}), \qquad H^{(0)} \leftarrow \mathrm{Read}(H^{(0)}, \tilde{L}).$$

This consolidation step uses only support features before label injection, and therefore does not expose query tokens to any ground-truth query information.

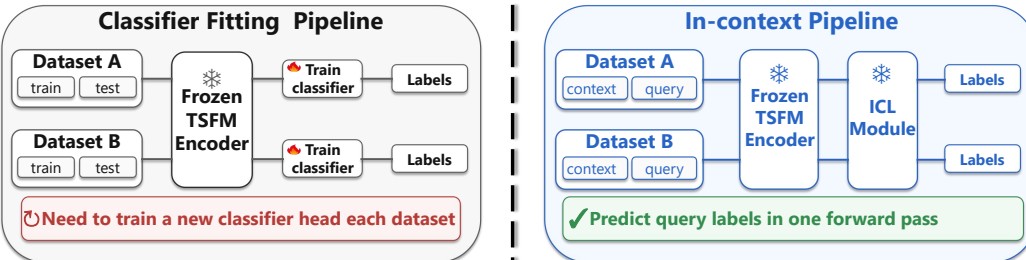

*Figure 1.* **From classifier fitting to in-context inference.** The standard frozen-backbone classifier-fitting pipeline extracts representations and fits a new task-specific classifier for each downstream dataset. In contrast, TIC-FM treats labeled examples as support context and predicts query labels through a fixed in-context inference module, avoiding deployment-time classifier fitting.

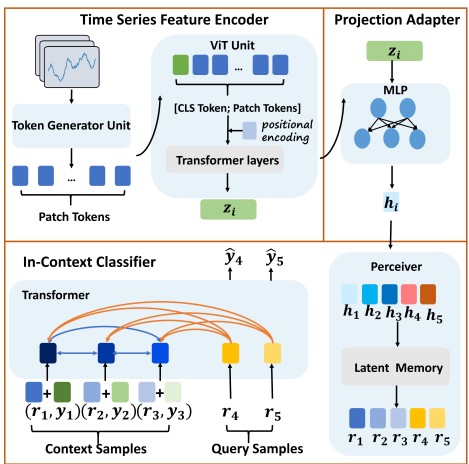

*Figure 2.* **Overview of TIC-FM.**

Second, task supervision is introduced by injecting label embeddings only into the support slice. Let $E_y : \{0, \ldots, C_{\max} - 1\} \to \mathbb{R}^d$ be a learnable label embedding map. We update $H^{(0)}_{1:N_{\text{tr}}} \leftarrow H^{(0)}_{1:N_{\text{tr}}} + E_y(y_{\text{tr}})$, $H^{(0)}_{N_{\text{tr}}+1:N}$ unchanged. Thus, labeled support examples serve as in-context prompts, while query tokens remain unlabeled.

Finally, the label-injected sequence is processed by a Transformer encoder with a split attention mask $\mathcal{M}(N_{\text{tr}})$. Context tokens attend only within the support set, and query tokens attend to support tokens, preventing query-query information leakage. The resulting representations are decoded into class logits:

$$H^{(L)} = \text{Transformer}\Big(H^{(0)}; \mathcal{M}(N_{\text{tr}})\Big),$$

$$\ell = D_\omega\Big(\text{LN}(H^{(L)})\Big).$$

where $D_\omega$ is a token-wise MLP producing $C_{\max}$ logits. For a task with $K$ active classes, we restrict the query logits to the active class indices $\mathcal{S}_{\text{act}}$ and apply a temperature-scaled softmax: $\hat{P}_{\text{te}} = \text{Softmax}(\ell_{N_{\text{tr}}+1:N}, \mathcal{S}_{\text{act}}/T_{\text{temp}}) \in \mathbb{R}^{N_{\text{te}} \times K}$. Full architectural details are provided in Appendix B.

**Pretraining and Inference.** For pretraining, we adopt a stage-wise training strategy tailored to each component's role. We first use Cauker (Xie et al., 2025a) to generate 100K synthetic time series samples and train the feature encoder $F_\psi$ for 100 epochs with a contrastive objective. We then pretrain the in-context classifier $G_\theta$ on synthetic data drawn from a structural causal model prior (Bouadi et al., 2025). Finally, using the UCR training splits, we freeze $F_\psi$ and $G_\theta$ and train only the projection adapter $g_\phi$ for 5 epochs with cross-entropy. No information from UCR test splits is used at any stage. Alternatively, $g_\phi$ can also be trained purely on SCM data, yielding a fully synthetic pre-trained TIC-FM, further details are provided in Appendix C.1. For inference, we apply an ensembling strategy to improve robustness. Given a context set $\mathcal{D}_{\text{tr}}$ and a query set $\mathcal{D}_{\text{te}}$ with $K$ classes, we form $X_{\text{all}} = [X_{\text{tr}}; X_{\text{te}}]$ and evaluate $M$ ensemble members. Each member is defined by a cyclic label permutation applied only to the context labels, $\pi_{o_m}(y) = (y + o_m) \bmod K$. We run $G_\theta$ to obtain query logits $\ell_m(\cdot)$ under offset $o_m$, map them back to the original label space via $\pi_{o_m}^{-1}$, and aggregate across members:

$$\hat{p}(y \mid x) = \text{Softmax}\left(\frac{1}{T_{\text{temp}}} \cdot \frac{1}{M} \sum_{m=1}^{M} \pi_{o_m}^{-1}(\ell_m(x; o_m))\right).$$

Additionally, when $K$ exceeds $C_{\max}$, we adopt a hierarchical class-extension strategy (Qu et al., 2025) by constructing a balanced class-partition tree that recursively decomposes the $K$-way task into subproblems, each involving at most $C_{\max}$ classes, and combines group and within-group predictions via the law of total probability. Further inference details are provided in Appendix C.2.

While our main contribution is empirical and methodological, Appendix D gives an expressivity view of support-conditioned ICL as an approximation to continuous classifier-fitting maps.

## 4. Experiments

**Setup.** We evaluate TIC-FM on the UCR Time Series Archive (Dau et al., 2019), using the official train/test splits

*Table 1.* **Classification Accuracy and Average Rank on full UCR Datasets.** "Fit?" indicates deployment-time task-specific classifier fitting. The best results are in **bold**, and the second best results are underlined.

| METHOD | FIT? | AVG ACC | AVG RANK |
|---|---|---|---|
| MOMENT + RF | YES | 77.51% | 5.32 |
| MOMENT + SVM | YES | 77.98% | 4.44 |
| MOMENT + MLP | YES | 44.51% | 11.42 |
| MOMENT + KNN | NO | 75.72% | 6.65 |
| MOMENT + NC | NO | 66.06% | 9.34 |
| MANTIS + RF | YES | 78.67% | 4.82 |
| MANTIS + SVM | YES | 79.06% | 4.44 |
| MANTIS + MLP | YES | 63.53% | 9.52 |
| MANTIS + KNN | NO | 77.07% | 5.80 |
| MANTIS + NC | NO | 70.53% | 7.84 |
| TIC-FM (SYN.) | NO | 79.75% | 4.42 |
| **TIC-FM** | NO | **80.01%** | **4.00** |

unless otherwise stated. We compare against two representative TSFMs, Mantis (Feofanov et al., 2025) and MOMENT (Goswami et al., 2024), paired with trained classifiers (RF, SVM, MLP) and training-free classifiers (kNN, nearest centroid). We report two variants: **TIC-FM**, which uses a frozen pretrained encoder and in-context classifier with a lightweight adapter calibrated on UCR training splits, and **TIC-FM (Syn.)**, a fully synthetic transfer variant using no UCR training labels for calibration. Accuracy is the primary metric, and all results are averaged over five random seeds. Full protocols, per-dataset results, and implementation details are provided in Appendix F.

**Main results.** Table 1 summarizes the standard UCR results. Under the official train/test split, TIC-FM achieves the best average accuracy and average rank while avoiding deployment-time classifier fitting. The comparison also reveals substantial classifier sensitivity for the same frozen backbone, e.g., Mantis and MOMENT yield noticeably different results when paired with SVM, RF, MLP, kNN, or nearest-centroid classifiers. This supports our motivation that frozen-backbone evaluation can conflate representation quality with downstream classifier choice.

**Robustness under extreme label scarcity.** We further evaluate TIC-FM in an extreme low-shot regime, where only 10% or 15% of each dataset is revealed as labeled context. Because Mantis and MOMENT may have been pretrained using UCR training splits, we avoid potential leakage by discarding the official UCR training split in this low-shot protocol. Instead, each episode is constructed solely from the official test split: a stratified subset is used as the labeled context set, with at least one example per class whenever feasible, and the remaining samples are used as queries. As shown in Table 2, TIC-FM achieves the highest average

accuracy under both label fractions. TIC-FM (Syn.) also outperforms all frozen-backbone baselines, indicating that synthetic-only adapter training already provides strong low-shot transfer. Under severe label scarcity, classifier-fitting baselines vary substantially across classifier families: parametric heads such as one-layer MLPs may underfit or overfit, while SVM, RF, kNN, and nearest-centroid classifiers are more stable but rely on fixed optimization or geometric biases. In contrast, TIC-FM directly models support-query interactions through in-context inference without per-dataset classifier fitting, yielding stronger low-label performance.

*Table 2.* **Extreme low-shot results.** "Fit?" indicates deployment-time task-specific classifier fitting. Results are average accuracy over the full UCR archive. Best is **bold**; second best is underlined.

| Method | Fit? | 10% | 15% |
|---|---|---|---|
| MOMENT + RF | Yes | 69.19% | 71.76% |
| MOMENT + SVM | Yes | 70.13% | 72.88% |
| MOMENT + MLP | Yes | 43.64% | 43.84% |
| MOMENT + kNN | No | 68.93% | 71.99% |
| MOMENT + NC | No | 63.41% | 64.85% |
| Mantis + RF | Yes | 69.57% | 72.14% |
| Mantis + SVM | Yes | 71.17% | 72.91% |
| Mantis + MLP | Yes | 56.77% | 59.67% |
| Mantis + kNN | No | 70.62% | 72.80% |
| Mantis + NC | No | 66.07% | 67.45% |
| TIC-FM (Syn.) | No | 71.48% | 73.91% |
| **TIC-FM** | No | **72.30%** | **74.59%** |

Appendix E reports broader UCR comparisons, scaling studies, ablations, and efficiency analysis. TIC-FM achieves the best average accuracy among TiCT, TimesNet, classical classifiers, and additional TSFM backbones, while improving with more labeled context, gaining mainly from support-conditioned in-context inference, and reducing end-to-end latency despite higher peak memory (Table 3, Figures 3–5, and Table 4).

## 5. Conclusion

In this work, we revisit the frozen-backbone classifier-fitting pipeline for TSFM-based classification and show that it is not truly training-free and remains sensitive to downstream classifier choice. We propose TIC-FM, a support-conditioned in-context classification framework that predicts query labels from labeled context examples without fitting task-specific classifiers at deployment time. This formulation is well-suited to settings where parameter updates are impractical, including on-device deployment and privacy-sensitive applications. We hope TIC-FM encourages a broader shift from classifier fitting toward in-context inference for time series classification foundation models.

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

# A. Related Work

**Time series foundation models (TSFMs).** TSFMs leverage large-scale pretraining to learn transferable representations that can be adapted to new domains, mirroring the foundation model paradigm established by large language models (LLMs) and vision models (Liang et al., 2024). Recent progress has been especially striking in forecasting, where large pretrained models support standardized deployment modes, including zero-shot evaluation, few-shot adaptation, and task-specific fine-tuning (Ansari et al., 2024; 2025; Das et al., 2023; Cohen et al., 2024; Woo et al., 2024; Auer et al., 2025b; Moroshan et al., 2025; Rasul et al., 2024). In contrast, time series classification with foundation models is still commonly operationalized as frozen feature extraction followed by a trained downstream classifier, which requires task-specific classifier fitting when the model is transferred to a new labeled dataset (Feofanov et al., 2025; Lin et al., 2023; Zhang et al., 2025; Xie et al., 2025a; Auer et al., 2025a; Gao et al., 2024; Zhou et al., 2023). Among classification-oriented TSFMs, models such as MOMENT (Goswami et al., 2024) have popularized the embedding-classifier paradigm and are now widely used as backbones. However, a broadly applicable training-free inference framework for time series classification remains underexplored.

**In-context learning (ICL).** ICL, popularized by LLMs, refers to task adaptation at inference time by conditioning on a small set of input–output exemplars, without any parameter updates (Dong et al., 2024). Prevailing mechanistic accounts interpret ICL either as implicit Bayesian inference over latent task structure (Xie et al., 2021; Falck et al., 2024; Panwar et al., 2024) or as optimization-like dynamics implemented by the Transformer forward pass (Wies et al., 2023; Ahn et al., 2023; von Oswald et al., 2023; Mahankali et al., 2023; Xie et al., 2025b). In contrast to the rich literature on ICL in LLMs, the in-context capabilities of TSFMs remain far less explored. Moreover, in time series forecasting, the notion of "context" is often conflated with the historical lookback window rather than exemplar-based conditioning (Wu et al., 2022; Auer et al., 2025b; Lu et al., 2025; Faw et al., 2025). Time series classification, however, naturally matches the ICL paradigm: context examples can be represented as sequence–label pairs, enabling query labels to be predicted in a single forward pass conditioned on a labeled context set, without task-specific parameter updates. This sequence-label formulation also mirrors recent ICL approaches for tabular foundation models (Hollmann et al., 2022; Qu et al., 2025), suggesting an emerging unification across foundation models for structured data (Müller et al., 2024; Hoo et al., 2025; Xie et al., 2025a). Concurrently, Tokic et al. (2025) and Yeh et al. (2025) have also begun to investigate ICL for time series classification: the former focuses on an application-driven bearing-health setting, while TiCT (Yeh et al., 2025) is the closest general-purpose ICL classifier, using retrieval-based per-query inference. In contrast, TIC-FM operates at the episode level: it produces all query labels in a single forward pass without retrieval-based per-query prompting. Beyond architecture, we focus on the evaluation and deployment pipeline, showing that in-context inference offers a training-free alternative that avoids classifier-choice sensitivity, hyperparameter tuning, and per-dataset classifier fitting.

# B. TIC-FM Methodology Details

## B.1. Embedding-to-ICL Framework

Following the training-free ICL setting in Section 2, TIC-FM jointly processes the concatenated context and query sets and produces predictions for all query instances in parallel. A schematic overview of the architecture is presented in Figure 2. Specifically, TIC-FM comprises three components with a clear information-flow constraint. First, a time series encoder $F_\psi$ summarizes the within-sequence temporal structure into an instance embedding that supports cross-instance comparison. Next, a projection adapter $g_\phi$ maps these embeddings into the token space expected by the in-context Transformer. Finally, an in-context classifier $G_\theta$ performs inference by attending to the labeled context tokens and the unlabeled query tokens while enforcing a strict separation between the two sets, thereby enabling parallel prediction for all query samples.

## B.2. Time Series Feature Encoder

We use a time series encoder $F_\psi$ to map each input series $x \in \mathbb{R}^T$ to an embedding $z = F_\psi(x) \in \mathbb{R}^q$. The encoder first employs a token generator unit to transform the raw series into a sequence of patch tokens that capture local temporal dynamics and patch-level statistics, and then applies a ViT to aggregate information across the token sequence. The final embedding $z$ is obtained from a designated pooling token (i.e., a classification token), which attends to all patch tokens to summarize their information and thus yields a semantically rich embedding of $x$.

**Token Generator Unit.** Drawing inspiration from Mantis (Feofanov et al., 2025), the token generator constructs a sequence of patch embeddings $U \in \mathbb{R}^{P \times q}$ that integrates both local dynamics and patch-level statistical properties. We combine local dynamics with patch-level statistics to improve invariance to shifts and to preserve discriminative cues when the

labeled context is limited. Specifically, we partition the input time series $x \in \mathbb{R}^T$ into $P$ non-overlapping patches of length $w = T/P$ (assuming $T$ is divisible by $P$). Let $x^{(p)} \in \mathbb{R}^w$ denote the $p$-th patch for $p = 1, \ldots, P$. We first compute the mean and standard deviation for each patch to capture patch-level statistical properties:

$$\mu_p = \mathrm{mean}(x^{(p)}) \in \mathbb{R}, \qquad \sigma_p = \mathrm{std}(x^{(p)}) \in \mathbb{R}.$$

These statistics are encoded via a learnable statistic encoder $\phi_{\mathrm{stat}}(\cdot) : s_p = \phi_{\mathrm{stat}}([\mu_p; \sigma_p]) \in \mathbb{R}^{d_s}$, where $[\cdot; \cdot]$ denotes concatenation and $d_s$ is the statistic-embedding dimension. To extract short-term temporal patterns, we construct two complementary views of the input: the raw series $x$ and its first-order difference $\Delta x$ (zero-padded to length $T$). We employ a shared local feature extractor $\phi_{\mathrm{loc}}(\cdot)$ to obtain patch-aligned dynamic representations:

$$V^{\mathrm{raw}} = \phi_{\mathrm{loc}}(x), \qquad V^{\mathrm{diff}} = \phi_{\mathrm{loc}}(\Delta x), \qquad V^{\mathrm{raw}}, V^{\mathrm{diff}} \in \mathbb{R}^{P \times d_\ell},$$

where $d_\ell$ denotes the dynamic feature dimension. In our implementation, $\phi_{\mathrm{loc}}$ is instantiated as a stack of convolutional layers followed by patch-wise pooling, ensuring alignment with the patch segmentation. Finally, we fuse the dynamic and statistical features via concatenation and a learnable projection $\phi_{\mathrm{tok}}$:

$$u_p = \phi_{\mathrm{tok}}\big([V_p^{\mathrm{diff}}; V_p^{\mathrm{raw}}; s_p]\big) \in \mathbb{R}^q, \qquad U = [u_1^\top; \ldots; u_P^\top] \in \mathbb{R}^{P \times q},$$

where $V_p^{\mathrm{diff}}$ and $V_p^{\mathrm{raw}}$ denote the $p$-th rows of their respective matrices. In this unit, patch tokens are constructed as the basic evidence units for context-based matching. We combine local dynamics with patch-level statistics to improve invariance to shifts and to preserve discriminative cues under limited labeled context.

**ViT Unit.** We adopt a ViT Unit (Dosovitskiy, 2020; Feofanov et al., 2025) to summarize the patch tokens because self-attention implements content-adaptive pooling, allowing the model to emphasize informative temporal segments that are most relevant for classification. This yields a compact and discriminative instance representation that is well suited for downstream in-context classification. Given the patch embedding sequence $U \in \mathbb{R}^{P \times q}$ produced by the token generator above, the ViT unit is used to aggregate information across patches and obtain a global representation. Specifically, we introduce a learnable classification token $u_{\mathrm{cls}} \in \mathbb{R}^{1 \times q}$ and prepend it to the patch tokens: $\hat{U}^{(0)} = [u_{\mathrm{cls}}; U] \in \mathbb{R}^{(P+1) \times q}$. We then add positional encodings $E_{\mathrm{pos}} \in \mathbb{R}^{(P+1) \times q}$ to inject patch order information, $\tilde{U}^{(0)} = \hat{U}^{(0)} + E_{\mathrm{pos}}$, and feed the resulting sequence into an $L$-layer Transformer:

$$\tilde{U}^{(L)} = \mathrm{Transformer}_\psi\big(\tilde{U}^{(0)}\big) \in \mathbb{R}^{(P+1) \times q}.$$

Finally, we take the output corresponding to the classification token as the instance embedding: $z = \tilde{U}_0^{(L)} \in \mathbb{R}^q$. In our implementation, the Transformer uses multi-head self-attention with $h$ heads and an MLP feed-forward block at each layer, matching the standard ViT design.

## B.3. Projection Adapter

The encoder embedding space $\mathbb{R}^q$ is not necessarily aligned with the token space expected by the in-context classifier. Since in-context inference is driven by attention weights and token-wise interactions, mismatched feature statistics can degrade conditioning on labeled context. We therefore learn a lightweight projection adapter $g_\phi$ to map instance embeddings into an ICL-compatible representation space. We implement $g_\phi$ as an MLP with Layer Normalization (Ba et al., 2016):

$$g_\phi(z) = W_2\, \sigma\big(W_1\, \mathrm{LN}(z)\big), W_1 \in \mathbb{R}^{d_h \times q}, W_2 \in \mathbb{R}^{d \times d_h},$$

where $\sigma$ is GELU (Hendrycks, 2016). The adapter is applied independently to each instance embedding, enabling efficient batch processing of dataset sequences. In our implementation, $d$ matches the internal model dimension of the ICL Transformer.

## B.4. In-Context Classifier

We denote the projected feature embeddings from the upstream encoder as $H_{\mathrm{tr}} \in \mathbb{R}^{N_{\mathrm{tr}} \times d}$ and $H_{\mathrm{te}} \in \mathbb{R}^{N_{\mathrm{te}} \times d}$. The In-Context Classifier, denoted as $G_\theta$, processes the concatenated sequence $H^{(0)} = [H_{\mathrm{tr}}; H_{\mathrm{te}}] \in \mathbb{R}^{N \times d}$, where $N = N_{\mathrm{tr}} + N_{\mathrm{te}}$. The computation consists of three stages: latent memory consolidation, supervision injection, and split-masked in-context reasoning.

**Latent Context Consolidation.** Before label injection, we incorporate a Perceiver-style latent memory module to consolidate global information from the support context. The mechanism maintains learnable latent queries $L \in \mathbb{R}^{M \times d}$ with $M \ll N_{\text{tr}}$, and applies a stack of cross-attention and feed-forward blocks. Concretely, it first writes information from the context tokens into the latents and then reads the updated latents back to refine all tokens:

$$\tilde{L} = \text{Write}(L, H_{\text{tr}}), \qquad H^{(0)} \leftarrow \text{Read}\Big(H^{(0)}, \tilde{L}\Big).$$

Crucially, the write operation only accesses context features without labels, and the subsequent read refines both context and query representations via latent summary, thereby remaining label-leak safe.

**Task-Aware Label Injection.** We encode class labels using a one-hot projection $E_y : \{0, \ldots, C_{\max}-1\} \rightarrow \mathbb{R}^d$ implemented by a linear layer over one-hot inputs. Labels are injected additively only into the context slice:

$$H_{1:N_{\text{tr}}}^{(0)} \leftarrow H_{1:N_{\text{tr}}}^{(0)} + E_y(y_{\text{tr}}), \qquad H_{N_{\text{tr}}+1:N}^{(0)} \text{ unchanged},$$

so that ground-truth labels act as in-context prompts while query tokens remain unlabeled.

**Split-masked In-Context Reasoning.** The label-injected sequence is processed by an $L$-layer Transformer encoder. To enforce a strict separation between context and queries, we apply a split attention mask $\mathcal{M}(N_{\text{tr}})$ defined by the boundary between the context and the query tokens. Under this mask, context tokens attend only within the context set, and each query token attends only to the context tokens. This masking prevents information leakage between query tokens. The encoder outputs are then normalized and decoded into class logits:

$$H^{(L)} = \text{Transformer}\big(H^{(0)}; \mathcal{M}(N_{\text{tr}})\big), \qquad \ell = D_\omega\big(\text{LN}(H^{(L)})\big),$$

where $D_\omega$ is a token-wise two-layer MLP producing $C_{\max}$ logits. For prediction, the decoder produces per-token logits $\ell \in \mathbb{R}^{N \times C_{\max}}$. Let $\mathcal{S}_{\text{act}} \subseteq \{0, \ldots, C_{\max} - 1\}$ be the index set of the $K$ classes appearing in the context labels (after task-wise re-indexing). We take the query slice restricted to the active classes, $\ell_{\text{te}} := \ell_{N_{\text{tr}}+1:N, \, \mathcal{S}_{\text{act}}} \in \mathbb{R}^{N_{\text{te}} \times K}$ with $|\mathcal{S}_{\text{act}}| = K$. We then apply a row-wise temperature-scaled softmax, $\hat{P}_{\text{te}} = \text{Softmax}(\ell_{\text{te}}/T_{\text{temp}}) \in \mathbb{R}^{N_{\text{te}} \times K}$, where the softmax is applied over the class dimension.

### B.5. Pretraining and Inference

For pretraining, we adopt a stage-wise training strategy tailored to each component's role. We first use Cauker (Xie et al., 2025a) to generate 100K synthetic time series samples and train the feature encoder $F_\psi$ for 100 epochs with a contrastive objective. We then pretrain the in-context classifier $G_\theta$ on synthetic data drawn from a structural causal model prior (Bouadi et al., 2025). Finally, using the UCR training splits, we freeze $F_\psi$ and $G_\theta$ and train only the projection adapter $g_\phi$ for 5 epochs with cross-entropy. No information from UCR test splits is used at any stage. Alternatively, $g_\phi$ can also be trained purely on SCM data, yielding a fully synthetic pre-trained TIC-FM, further details are provided in Appendix C.1. For inference, we apply an ensembling strategy to improve robustness. Given a context set $\mathcal{D}_{\text{tr}}$ and a query set $\mathcal{D}_{\text{te}}$ with $K$ classes, we form $X_{\text{all}} = [X_{\text{tr}}; X_{\text{te}}]$ and evaluate $M$ ensemble members. Each member is defined by a cyclic label permutation applied only to the context labels, $\pi_{o_m}(y) = (y + o_m) \mod K$. We run $G_\theta$ to obtain query logits $\ell_m(\cdot)$ under offset $o_m$, map them back to the original label space via $\pi_{o_m}^{-1}$, and aggregate across members:

$$\hat{p}(y \mid x) = \text{Softmax}\left(\frac{1}{T_{\text{temp}}} \cdot \frac{1}{M} \sum_{m=1}^{M} \pi_{o_m}^{-1}(\ell_m(x; o_m))\right).$$

Additionally, when $K$ exceeds $C_{\max}$, we adopt a hierarchical class-extension strategy (Qu et al., 2025) by constructing a balanced class-partition tree that recursively decomposes the $K$-way task into subproblems, each involving at most $C_{\max}$ classes, and combines group and within-group predictions via the law of total probability. Further inference details are provided in Appendix C.2.

While our main contribution is empirical and methodological, Appendix D provides an explanatory expressivity perspective showing that support-conditioned in-context models can approximate continuous classifier-fitting maps under explicit assumptions.

# C. Details of Pretraining and Inference

## C.1. Pretraining

We first pretrain the time series feature encoder $F_\psi$ (Section B.2) for 100 epochs on 100K synthetic time series generated by Cauker (Xie et al., 2025a) using a contrastive objective. This pretraining encourages $F_\psi$ to learn discriminative representations by maximizing agreement between two stochastically augmented views of the same instance while reducing similarity across different instances.

Formally, let $\mathcal{B} = \{x_i\}_{i=1}^B$ denote a mini-batch of $B$ synthetic time series. For each $x_i$, we sample two augmentation operators $\mathcal{T}_1, \mathcal{T}_2 \sim \mathcal{T}$ (e.g., random cropping and resizing) to obtain two correlated views $\tilde{x}_{i,1} = \mathcal{T}_1(x_i)$ and $\tilde{x}_{i,2} = \mathcal{T}_2(x_i)$. The two views are encoded by $F_\psi$ and mapped to a contrastive latent space by a projection head $g_\phi(\cdot)$, yielding

$$z_{i,1} = g_\phi\big(F_\psi(\tilde{x}_{i,1})\big), \qquad z_{i,2} = g_\phi\big(F_\psi(\tilde{x}_{i,2})\big).$$

In our implementation, $g_\phi$ is instantiated as a Layer Normalization followed by a linear projection.

Following (Oord et al., 2018; He et al., 2020), we adopt a one-way in-batch InfoNCE loss, treating $z_{i,1}$ as queries and $z_{i,2}$ as keys. We first $\ell_2$-normalize the embeddings, $\bar{z}_{i,1} = z_{i,1}/\|z_{i,1}\|_2$ and $\bar{z}_{i,2} = z_{i,2}/\|z_{i,2}\|_2$, and compute the pairwise similarity logits

$$s_{ij} = \frac{\bar{z}_{i,1}^\top \bar{z}_{j,2}}{\tau}, \qquad \tau > 0,$$

forming a $B \times B$ logit matrix $S = [s_{ij}]$. For each query index $i$, the positive key is the matched index $j = i$, and all $j \neq i$ serve as in-batch negatives. The resulting contrastive loss is the cross-entropy over the in-batch keys:

$$\mathcal{L}_{\text{con}} = \frac{1}{B} \sum_{i=1}^B \left( -\log \frac{\exp(s_{ii})}{\sum_{j=1}^B \exp(s_{ij})} \right),$$

equivalently $\mathcal{L}_{\text{con}} = \text{CE}(S, [1, 2, \ldots, B])$, where the target indices correspond to the diagonal alignment. Unless otherwise stated, we use $\tau = 0.1$.

Regarding the optimization of the projection adapter $g_\phi$ (Section B.3) and the in-context classifier $G_\theta$ (Section B.4), specifically when the adapter is instantiated as an MLP, we employ a two-stage training protocol. In the first stage, we pretrain the in-context classifier on synthetic datasets generated from structural causal models (SCMs) for 27050 steps, following the Orion-MSP methodology (Bouadi et al., 2025). Subsequently, in the second stage, we freeze both the time series feature encoder and the pretrained in-context classifier. The projection adapter is then trained for 5 epochs on the UCR training splits using a standard cross-entropy objective. Specifically, we adopt an episodic training paradigm: for each iteration, we construct a classification task by sampling a labeled context set and a batch of query examples from a dataset's training split. The model predicts query labels conditioned on the context, and the loss is computed on these predictions. All gradients are backpropagated exclusively to the adapter parameters. Crucially, no samples from the UCR test sets are accessed during this process, thereby preventing any data leakage in our experiments.

To accommodate varying feature dimensions, we introduce RowMixerLite, a Transformer-based projection adapter that treats the feature dimension as a token sequence, enabling a unified interface for downstream in-context classification across heterogeneous time-series embedding dimensionalities. Concretely, given an input representation, RowMixerLite partitions the feature axis into non-overlapping patches of size 8 and applies a shared patch projection to map each patch into a token of dimension $d_{\text{model}}=128$. The resulting patch tokens are augmented with a small set of learnable special tokens, consisting of 4 class tokens and 2 global tokens, each of dimension 128. These tokens provide dedicated aggregation slots that summarize patch-level evidence and improve the stability of the subsequent pooling operation. We then apply a 3 layer Transformer encoder over the token sequence, using 8 attention heads, feed-forward dimension 256, pre-norm, and dropout 0.0. The output tokens are finally arranged into 4 class-specific embeddings, which are concatenated to match the token dimensionality expected by the in-context classifier, yielding $d_{\text{icl}}=512$.

Regarding the optimization of RowMixerLite and the in-context classifier, we jointly pretrain them end-to-end for 15750 steps on SCM-based synthetic data generated by Orion-MSP, which is designed to mimic diverse input-representation distributions. To improve robustness and reduce sensitivity to feature ordering, we apply feature shuffling with probability 0.25 during training. After pretraining, we integrate the RowMixerLite adapter with our time series encoder $F_\psi$ and

the in-context classifier $G_\theta$ for time series classification. This fully synthetic model supports multivariate time series classification and achieves competitive performance on UCR, reaching **79.75%** average accuracy. These results suggest that the TIC-FM framework can be trained entirely on synthetic data, without any real datasets, while remaining competitive.

## C.2. Inference Details

### C.2.1. HIERARCHICAL CLASS EXTENSION FOR $K > C_{\max}$

To address scenarios where the total number of classes $K$ exceeds the model's architectural limit $C_{\max}$ (constrained by the pre-defined label embedding dimension or context window), we implement a hierarchical class-extension strategy following (Qu et al., 2025). Instead of simple truncation, this method dynamically constructs a classification tree $\mathcal{T}$ derived from the labeled support set $\mathcal{D}_{\mathrm{tr}}$, allowing TIC-FM to perform inference over an arbitrary number of classes without parameter updates.

**Tree Construction (Fit Phase).** The construction process proceeds recursively starting from the root. Let $\mathcal{D}_\mathcal{N} = \{(x_i, y_i)\}$ denote the subset of support samples reaching node $\mathcal{N}$, and $\mathcal{Y}_\mathcal{N}$ be the set of unique classes present in $\mathcal{D}_\mathcal{N}$. The tree is built based on the following logic:

1. **Leaf Condition:** If $|\mathcal{Y}_\mathcal{N}| \leq C_{\max}$, the node $\mathcal{N}$ is designated as a *leaf*. It directly stores $\mathcal{D}_\mathcal{N}$ as its local in-context demonstrations for standard inference.

2. **Internal Node Splitting:** If $|\mathcal{Y}_\mathcal{N}| > C_{\max}$, the node becomes an internal router. We set
$$G_\mathcal{N} = \min\{|\mathcal{Y}_\mathcal{N}|, C_{\max}\},$$
and partition the classes $\mathcal{Y}_\mathcal{N}$ into $G_\mathcal{N}$ disjoint groups $\{\mathcal{G}_1, \ldots, \mathcal{G}_{G_\mathcal{N}}\}$ with balanced sizes. This ensures that the routing problem at each internal node has at most $C_{\max}$ group labels.

3. **Label Coarsening (Meta-Task Construction):** We construct a coarse-grained classification task for the internal node. The original labels $y_i$ in $\mathcal{D}_\mathcal{N}$ are mapped to their corresponding group indices
$$g(y_i) \in \{0, \ldots, G_\mathcal{N} - 1\}.$$
This forms a meta-support set $\mathcal{D}'_\mathcal{N} = \{(x_i, g(y_i))\}$, which serves as the context for deciding which branch to traverse.

4. **Recursion:** We instantiate $G_\mathcal{N}$ child nodes, where the $j$-th child is recursively fitted using only the subset of data belonging to group $\mathcal{G}_j$. If a child group still contains more than $C_{\max}$ classes, the same splitting rule is applied recursively.

**Recursive Inference (Predict Phase).** During inference, a query sample $x_{\mathrm{te}}$ traverses the tree from the root. The probability of a final class $y$ is computed via the chain rule of probability along the path from the root to the leaf containing $y$. For an internal node $\mathcal{N}$, the model behaves as a router, predicting the probability distribution over groups $P(g \mid x_{\mathrm{te}}; \mathcal{D}'_\mathcal{N})$ using the meta-support set. Algorithm 1 formalizes this recursive probability aggregation.

### C.2.2. TEST-TIME ENSEMBLING VIA CYCLIC LABEL PERMUTATIONS

**Ensemble members.** Let $K$ be the number of classes in the current dataset. We construct an ensemble of $M = n_{\mathrm{est}}$ members, each parameterized by a cyclic label-shift offset $o_m \in \{0, \ldots, K - 1\}$. In practice, we generate offsets by shuffling $\{0, \ldots, K - 1\}$ and cycling through the list if $M > K$.

**Cyclic label permutation and aggregation.** For member $m$ with offset $o_m$, we apply a cyclic permutation *only* to the support labels:
$$\tilde{y}_{\mathrm{tr}} = \pi_{o_m}(y_{\mathrm{tr}}), \qquad \pi_{o_m}(y) = (y + o_m) \bmod K.$$
We then run the ICL model using $(X_{\mathrm{all}}, \tilde{y}_{\mathrm{tr}})$ to obtain the query logits $\ell_m \in \mathbb{R}^{N_{\mathrm{te}} \times K}$ in the permuted label space. To aggregate predictions across members, we map logits back to the original label space via the inverse permutation $\pi_{o_m}^{-1}$ (implemented as a circular shift along the class dimension) and average:
$$\bar{\ell} = \frac{1}{M} \sum_{m=1}^{M} \pi_{o_m}^{-1}(\ell_m).$$

---

**Algorithm 1** Hierarchical In-Context Inference
___
 1: **Input:** Query sample $x$, current node $\mathcal{N}$, model $\Phi$
 2: **Output:** Probability distribution over classes in $\mathcal{Y}_\mathcal{N}$
 3: **if** $\mathcal{N}$ is a leaf **then**
 4:     {Standard ICL inference with local support set}
 5:     Let $(\mathbf{R}, \mathbf{y})$ be the support set stored in $\mathcal{N}$
 6:     Return $\Phi(x \mid \mathbf{R}, \mathbf{y})$
 7: **else**
 8:     {Router step: predict group probabilities}
 9:     Let $(\mathbf{R}, \mathbf{g})$ be the meta-support set $\mathcal{D}'_\mathcal{N}$
10:     $\mathbf{P}_{\text{group}} \leftarrow \Phi(x \mid \mathbf{R}, \mathbf{g})$                                                    $\triangleright$ Shape: $[G_\mathcal{N}]$
11:     Initialize $\mathbf{P}_{\text{final}}$ as zeros over $\mathcal{Y}_\mathcal{N}$
12:     **for** each group index $j \in \{0, \ldots, G_\mathcal{N} - 1\}$ **do**
13:         Let $\mathcal{C}_j$ be the $j$-th child node
14:         $\mathbf{P}_{\text{child}} \leftarrow \text{HIERARCHICALPREDICT}(x, \mathcal{C}_j, \Phi)$             $\triangleright$ Distribution over $\mathcal{Y}_{\mathcal{C}_j}$
15:         {Accumulate probability mass: $P(y) = P(y \mid g = j)P(g = j)$}
16:         $\mathbf{P}_{\text{final}}[\mathcal{Y}_{\mathcal{C}_j}] \leftarrow \mathbf{P}_{\text{child}} \times \mathbf{P}_{\text{group}}[j]$
17:     **end for**
18:     Return $\mathbf{P}_{\text{final}}$
19: **end if**

---

# D. A Theoretical Perspective on In-Context Inference

## D.1. ICL Can Approximate Continuous Classifier-Fitting Pipelines

**Assumption D.1** (Bounded input domain and symmetry)**.** We fix maximum sizes $n_{\text{tr}} \leq N$ and $n_{\text{te}} \leq M$ and pad context–query input accordingly. We assume embeddings lie in a compact set and consider the resulting compact padded input domain $\mathcal{X}$. The induced score prediction map is permutation-invariant with respect to the order of training examples and equivariant in test examples.

**Proposition D.2** (Approximation of continuous classifier-fitting maps by ICL)**.** *Let* Train *be any classifier-training procedure that maps a training set to a classifier* $h_\tau : \mathbb{R}^q \to \mathbb{R}^K$. *Let the corresponding* score *map be* $F(\mathcal{D}_{\text{tr}}, \{z_j^{\text{te}}\}_{j=1}^{n_{\text{te}}}) \triangleq \{h_\tau(z_j^{\text{te}})\}_{j=1}^{n_{\text{te}}} \in (\mathbb{R}^K)^{n_{\text{te}}}$ *with* $h_\tau = \text{Train}(\mathcal{D}_{\text{tr}})$. *Assume the score map* $F$ *satisfies Assumption D.1 and is continuous on* $\mathcal{X}$.

*Then for any* $\varepsilon > 0$, *there exists an in-context classifier* $G_\theta$ *operating on the prompt tokens* $\{r(z_i^{\text{tr}}, y_i^{\text{tr}})\}_{i=1}^{n_{\text{tr}}} \cup \{r(z_j^{\text{te}}, \perp)\}_{j=1}^{n_{\text{te}}}$ *such that, on* $\mathcal{X}$,

$$\sup_{(U,V) \in \mathcal{X}} \max_{1 \leq j \leq n_{\text{te}}} \left\| G_\theta(U, V)_j - h_\tau(z_j^{\text{te}}) \right\|_\infty \leq \varepsilon.$$

In particular, this implies the following label-level guarantee under a uniform margin condition.

**Proof sketch.** View the trained-classifier pipeline as a continuous map that takes a set of labeled training tokens and a set of test tokens and returns test-time scores. Because the map is invariant to permutations of the training set, we can uniformly approximate each output coordinate by a polynomial that is symmetric in the training tokens. Any symmetric polynomial in the training tokens can be rewritten as a function of finitely many aggregated features of the form $\sum_i \phi(u_i)$ (Zaheer et al., 2017). A transformer-style in-context model can implement this computation by pooling token-wise features from training tokens into a summary representation using attention, and broadcasting the summary to each test token and applying a token-wise MLP. Combining these steps yields an in-context classifier whose test scores uniformly approximate those produced by the trained classifier. More details are available in Section D.3

**Discussion.** Proposition D.2 shows that, under the stated compactness, continuity, symmetry, and query-wise separability assumptions, a sufficiently expressive in-context model can uniformly approximate the score map induced by a frozen-backbone classifier-fitting pipeline.

## D.2. TIC-FM Emulates Gradient-Based Classifier Training

We provide a mechanistic justification for why a training-free in-context classifier can still behave like a learning algorithm: the forward pass can emulate gradient-style updates in its activations. This viewpoint is well-established in recent analyses of in-context learning in linear-attention models (von Oswald et al., 2023; Zhang et al., 2024; Xie et al., 2025b). We stress that this correspondence relies on idealized components, in particular linear attention, and should be read as an approximate analogy rather than a strict equivalence for the softmax Transformer used in TIC-FM.

**Proposition D.3** (In-context emulation of one gradient step). *Consider a scalar linear classifier trained on embeddings by one step of gradient descent on the unnormalized squared loss*

$$\ell(W) = \frac{1}{2} \sum_{i=1}^{n_{\text{tr}}} \left( W z_i^{\text{tr}} - y_i^{\text{tr}} \right)^2, \qquad W \in \mathbb{R}^{1 \times q},$$

*with initialization $W^{(0)} = 0$ and step size $\eta$. Then there exists an idealized TIC-FM instance whose in-context module $G_\theta$ is a linear-attention block such that, for every test token $r(z_j^{\text{te}}, \perp)$, the output scalar equals the one-step GD prediction:*

$$\left( G_\theta(\cdot) \right)_j^{(y)} = W^{(1)} z_j^{\text{te}}, \qquad j = 1, \dots, n_{\text{te}}.$$

*For the mean squared loss, the same construction holds when the support size is fixed, since the factor $1/n_{\text{tr}}$ can be absorbed into the fixed step size or attention scaling.*

**Proof sketch.** Let

$$e_i^{(t)} = W^{(t)} z_i^{\text{tr}} - y_i^{\text{tr}}$$

be the training residual. One gradient step on the unnormalized objective gives

$$W^{(t+1)} = W^{(t)} - \eta \sum_{i=1}^{n_{\text{tr}}} e_i^{(t)} (z_i^{\text{tr}})^\top.$$

Multiplying by a query embedding $z_j^{\text{te}}$ yields the prediction-space update

$$\hat{y}_j^{(t+1)} = \hat{y}_j^{(t)} - \eta \sum_{i=1}^{n_{\text{tr}}} e_i^{(t)} \left\langle z_i^{\text{tr}}, z_j^{\text{te}} \right\rangle.$$

A linear-attention block can implement this update by using the query–key product to compute $\langle z_i^{\text{tr}}, z_j^{\text{te}} \rangle$ and using the value projection to carry the residuals $e_i^{(t)}$ stored in dedicated token coordinates. The resulting unnormalized attention aggregation matches the summation in the gradient update and writes the update into the query label slot. A full derivation is deferred to Appendix D.4.

## D.3. Proof of Proposition D.2

We prove that an in-context classifier can uniformly approximate the trained-classifier score map on the compact padded-and-masked domain $\mathcal{X}$. By Assumption and the continuity of $r(\cdot, \cdot)$, the induced padded-and-masked prompt-token blocks $(U, V)$ lie in a compact set $\mathcal{X} \subset (\mathbb{R}^d)^N \times (\mathbb{R}^d)^M$.

### D.3.1. STEP 1: REDUCE TO APPROXIMATING A CONTINUOUS INVARIANT SCORE MAP

**Prompt tokenization.** Let $g_\phi : \mathbb{R}^q \to \mathbb{R}^d$ be the projection adapter and let $E_y : \{0, \dots, C_{\max} - 1\} \to \mathbb{R}^d$ be the label embedding. Extend it by $E_y(\perp) \triangleq 0 \in \mathbb{R}^d$ and define

$$r(z, y) \triangleq g_\phi(z) + E_y(y) \in \mathbb{R}^d.$$

In this proposition, the label space size equals the number of classes, i.e., $C_{\max} = K$.

Pad to fixed sizes $(N, M)$ and denote the resulting token blocks by $U = (u_1, \dots, u_N)$ and $V = (v_1, \dots, v_M)$, where $u_i = r(z_i^{\text{tr}}, y_i^{\text{tr}})$ and $v_j = r(z_j^{\text{te}}, \perp)$ for unmasked tokens. For padded positions, set $u_i = u_{\text{pad}}$ for $i > n_{\text{tr}}$ and $v_j = v_{\text{pad}}$ for $j > n_{\text{te}}$, with fixed $u_{\text{pad}}, v_{\text{pad}} \in \mathbb{R}^d$; these are ignored by the padding/masking convention.

The trained-classifier pipeline induces a score map

$$F(U,V) \triangleq \big(F_1(U,V),\dots,F_M(U,V)\big) \in (\mathbb{R}^K)^M,$$

where $F_j(U,V)$ equals $h_\tau(z_j^{\text{te}})$ for unmasked test tokens, with $h_\tau = \mathsf{Train}(\mathcal{D}_{\text{tr}})$. By assumption, $F$ is continuous on $\mathcal{X}$, permutation-invariant in the training blocks $U$, and permutation-equivariant in the test blocks $V$.

Thus it suffices to approximate $F$ uniformly on $\mathcal{X}$ by a model acting on the prompt tokens.

### D.3.2. STEP 2: SYMMETRIC POLYNOMIALS ARE DENSE FOR CONTINUOUS TRAINING-INVARIANT MAPS (SANNAI ET AL., 2019)

We first approximate each scalar coordinate of $F$ by a polynomial that is symmetric in the training blocks. Let $F_{j,k}(U,V)$ be the $k$-th coordinate of $F_j(U,V)$.

**Lemma D.4** (Density of polynomials symmetric in training blocks)**.** *Let $\mathcal{X} \subset (\mathbb{R}^d)^N \times (\mathbb{R}^d)^M$ be compact. For any continuous function $h : \mathcal{X} \to \mathbb{R}$ that is invariant under permutations of the $N$ training blocks, and any $\varepsilon > 0$, there exists a polynomial $p(U,V)$ such that (i) $p$ is symmetric in the training blocks $U$, and (ii) $\sup_{(U,V)\in\mathcal{X}} |h(U,V) - p(U,V)| \leq \varepsilon$.*

*Proof.* By Stone–Weierstrass, ordinary polynomials in all coordinates of $(U,V)$ are dense in $\mathcal{C}(\mathcal{X})$. Let $q(U,V)$ be a polynomial with $\sup_{\mathcal{X}} |h - q| \leq \varepsilon$. Define its symmetrization over the training blocks:

$$\mathrm{Sym}(q)(U,V) \triangleq \frac{1}{N!} \sum_{\pi \in S_N} q(\pi \cdot U, V).$$

Then $\mathrm{Sym}(q)$ is a polynomial symmetric in $U$. Since $h(\pi \cdot U, V) = h(U,V)$, we have $|h(U,V) - \mathrm{Sym}(q)(U,V)| \leq \sup_{\mathcal{X}} |h - q| \leq \varepsilon$ for all $(U,V) \in \mathcal{X}$. $\square$

Apply Lemma D.4 to each $F_{j,k}$ and take a union bound over finitely many coordinates. For any $\varepsilon > 0$, there exists a map $P(U,V) = (P_1,\dots,P_M) \in (\mathbb{R}^K)^M$ such that: (i) each $P_j$ is a polynomial in $(U,V)$ symmetric in $U$, and (ii)

$$\sup_{(U,V)\in\mathcal{X}} \max_{1\leq j\leq M} \|F_j(U,V) - P_j(U,V)\|_\infty \leq \varepsilon/2.$$

### D.3.3. STEP 3: SYMMETRIC POLYNOMIALS REDUCE TO SUMS OF ELEMENTWISE FEATURES (ZAHEER ET AL., 2017)

Since the trained classifier applies the same prediction rule to every query token, we approximate a single query-level score map rather than a family of position-indexed maps. Let

$$f(U,v) \in \mathbb{R}^K$$

denote the class-score vector produced for a query token $v$ conditioned on the support tokens $U = (u_1,\dots,u_N)$. This map is continuous on the compact padded domain and permutation-invariant in the support tokens.

For each output coordinate of $f$, by the symmetric-polynomial approximation argument in Step 2, there exists a polynomial $P(U,v) \in \mathbb{R}^K$ that is symmetric in $(u_1,\dots,u_N)$ and uniformly approximates $f(U,v)$. Since $P$ has finite total degree, there exists a finite set of monomials $\{m_{\alpha_r}\}_{r=1}^s$ on $\mathbb{R}^d$ such that $P$ can be written as a polynomial in the aggregated monomials $\sum_{i=1}^N m_{\alpha_r}(u_i)$, together with the query token $v$:

$$P(U,v) = \widetilde{P}\Big( \sum_{i=1}^N m_{\alpha_1}(u_i),\dots, \sum_{i=1}^N m_{\alpha_s}(u_i), v\Big).$$

Define

$$\phi(u) \triangleq \big(m_{\alpha_1}(u),\dots,m_{\alpha_s}(u)\big) \in \mathbb{R}^s, \qquad \rho(s,v) \triangleq \widetilde{P}(s,v) \in \mathbb{R}^K.$$

Then, for any query token $v$,

$$P(U,v) = \rho\Big( \sum_{i=1}^N \phi(u_i),\, v\Big),$$

which is a DeepSets-style form (Lee et al., 2019; Yun et al., 2019). Applying the same map to each query token yields the multi-query prediction

$$\big(P(U, v_1), \ldots, P(U, v_M)\big),$$

with $\phi$ and $\rho$ shared across query positions.

### D.3.4. Step 4: Realize the DeepSets computation with an in-context model

It remains to show that a transformer-like in-context model can implement the map

$$v_j \;\mapsto\; \rho\Big( \sum_{i=1}^N \phi(u_i),\; v_j \Big) \quad \text{uniformly on } \mathcal{X}.$$

**Lemma D.5** (Fixed-length pooling of training features). *Fix a maximum support size $N$ and pad each support set to $N$ slots with a padding token $u_{\mathrm{pad}}$. Suppose the token-wise feature map satisfies $\phi(u_{\mathrm{pad}}) = 0$. Then there exist parameters of a self-attention layer and a designated summary token $s$ such that, after applying a token-wise MLP implementing $\phi$ on support tokens, the summary token can represent*

$$s^{(1)} = \frac{1}{N} \sum_{i=1}^N \phi(u_i) = \frac{1}{N} \sum_{i=1}^{n_{\mathrm{tr}}} \phi(u_i).$$

*The unnormalized sum $\sum_{i=1}^{n_{\mathrm{tr}}} \phi(u_i)$ can be recovered by a following linear layer with the fixed scale factor $N$.*

*Proof.* Apply the token-wise MLP $\phi$ to all $N$ support slots and set $\phi(u_{\mathrm{pad}}) = 0$ for padded positions. Choose constant query and key projections so that the summary token attends uniformly to the fixed $N$ support slots. The attention output at the summary token is therefore

$$\frac{1}{N} \sum_{i=1}^N \phi(u_i) = \frac{1}{N} \sum_{i=1}^{n_{\mathrm{tr}}} \phi(u_i),$$

because padded slots contribute zero. Since $N$ is a fixed architectural constant, the factor $N$ can be absorbed into the subsequent linear layer, yielding the desired pooled sum. $\square$

**Lemma D.6** (Broadcast and apply a shared continuous map). *Assume the hidden dimension is large enough to contain two independent subspaces for query features and pooled support features. There exists a second attention layer in which each query token $v_j$ receives the pooled support representation $s^{(1)}$ in a subspace orthogonal to the one storing its own query representation. Consequently, after the residual addition, each query token contains both components, equivalently represented as $(v_j, s^{(1)})$. A subsequent shared token-wise MLP can uniformly approximate a continuous map*

$$(s^{(1)}, v) \mapsto \rho(s^{(1)}, v)$$

*on the compact domain. Applying this same map to every query token yields query-wise predictions for all queries.*

*Proof.* Use a fixed linear embedding to store each query representation in one hidden subspace and reserve another hidden subspace for the pooled support representation. Concretely, we may write a query token as

$$\bar{v}_j = (v_j, 0),$$

where the first block stores the query feature and the second block is reserved for the support summary. Set the attention logits so that each query token attends primarily to the summary token. Choose the attention value projection so that the attention output writes the pooled support representation only into the reserved subspace:

$$\mathrm{Attn}(\bar{v}_j, s^{(1)}) = (0, s^{(1)}).$$

After the residual addition, the query token becomes

$$\bar{v}_j + \mathrm{Attn}(\bar{v}_j, s^{(1)}) = (v_j, 0) + (0, s^{(1)}) = (v_j, s^{(1)}),$$

so the following token-wise MLP has independent access to both the query representation and the pooled support representation.

Since the trained classifier applies the same prediction rule to every query example after being fitted on the support set, the target can be written as a single shared query-level map

$$f(U, v),$$

where $U$ denotes the support tokens and $v$ denotes an arbitrary query token, rather than as a family of position-specific maps indexed by $j$. After the support set has been summarized by $s^{(1)}$, the desired approximation takes the shared form $\rho(s^{(1)}, v)$. By the universal approximation theorem on compact domains, a shared token-wise MLP can approximate this continuous map uniformly. Applying the same MLP independently to each query token gives the desired permutation-equivariant query predictions. $\square$

*Completion of the proof.* By Steps 2–3, there exists a shared DeepSets-form map $G^\star$ of the form

$$G_j^\star(U, V) = \rho\left(\sum_{i=1}^{N} \phi(u_i),\ v_j\right), \qquad j = 1, \ldots, M,$$

such that

$$\sup_{\mathcal{X}} \max_j \|F_j - G_j^\star\|_\infty \leq \varepsilon/2.$$

By Lemmas D.5–D.6, there exists an in-context model $G_\theta$ such that

$$\sup_{\mathcal{X}} \max_j \|G_j^\star - G_\theta(\cdot)_j\|_\infty \leq \varepsilon/2.$$

By the triangle inequality,

$$\sup_{\mathcal{X}} \max_j \|F_j - G_\theta(\cdot)_j\|_\infty \leq \varepsilon,$$

which proves the score-approximation claim in Proposition D.2. $\square$

**Corollary D.7** (Label matching under a uniform margin). *Let $h_\tau = \mathrm{Train}(\mathcal{D}_{\mathrm{tr}})$ and define the trained-classifier predicted label for each test embedding as*

$$y_j^\star = \arg\max_{k \in [K]} \left(h_\tau(z_j^{\mathrm{te}})\right)_k, \qquad j = 1, \ldots, n_{\mathrm{te}}.$$

*Assume the trained-classifier scores admit a* uniform margin *on $\mathcal{X}$, i.e., there exists $\gamma > 0$ such that for all $(\mathcal{D}_{\mathrm{tr}}, \{z_j^{\mathrm{te}}\}_{j=1}^{n_{\mathrm{te}}}) \in \mathcal{X}$ and all $j$,*

$$\left(h_\tau(z_j^{\mathrm{te}})\right)_{y_j^\star} - \max_{k \neq y_j^\star} \left(h_\tau(z_j^{\mathrm{te}})\right)_k \geq \gamma.$$

*If $G_\theta$ satisfies the score-approximation bound in Proposition D.2 with some $\varepsilon < \gamma/2$, and we define*

$$\hat{y}_j = \arg\max_{k \in [K]} \left(G_\theta(\cdot)_j\right)_k,$$

*then $\hat{y}_j = y_j^\star$ for all $j = 1, \ldots, n_{\mathrm{te}}$ on $\mathcal{X}$.*

**Interpretation.** The above result is an expressivity universal approximation statement in a continuous Euclidean token space. Proposition D.2 should be interpreted as an existence claim: there exist parameters $\theta$ such that an in-context model can approximate the target score map uniformly on $X$. It does not imply that a particular training algorithm will necessarily recover such parameters, nor does it provide unconditional guarantees for the realized optimization dynamics in practice.

### D.4. Proof of Proposition D.3

We prove that one linear-attention block can emulate one step of GD for a scalar linear head, following the prediction-space derivation in von Oswald et al. (2023); Zhang et al. (2024); Xie et al. (2025b).

D.4.1. STEP 1: GD IN PREDICTION SPACE

Let $\hat{y}^{(t)}(z) = W^{(t)}z$ with $W^{(0)} = 0$. Define training residuals $e_i^{(t)} = \hat{y}^{(t)}(z_i^{\mathrm{tr}}) - y_i^{\mathrm{tr}}$. One GD step gives

$$W^{(t+1)} = W^{(t)} - \eta \sum_{i=1}^{n_{\mathrm{tr}}} e_i^{(t)}(z_i^{\mathrm{tr}})^{\top}.$$

Multiplying by a query $z$ yields the prediction update

$$\hat{y}^{(t+1)}(z) = \hat{y}^{(t)}(z) - \eta \sum_{i=1}^{n_{\mathrm{tr}}} e_i^{(t)}\langle z_i^{\mathrm{tr}}, z\rangle.$$

Thus each step updates all query predictions using training residuals weighted by inner products.

D.4.2. STEP 2: ONE LINEAR-ATTENTION BLOCK IMPLEMENTS ONE PREDICTION UPDATE

Consider tokens whose first $q$ coordinates store the embedding $z$, and whose dedicated scalar slots store the ground-truth label $y$ and the current prediction $\hat{y}^{(t)}(z)$. For support tokens, both $y_i^{\mathrm{tr}}$ and $\hat{y}^{(t)}(z_i^{\mathrm{tr}})$ are available, so a linear value projection can form the residual

$$e_i^{(t)} = \hat{y}^{(t)}(z_i^{\mathrm{tr}}) - y_i^{\mathrm{tr}}.$$

For query tokens, the label slot is masked or set to a null value, and the block updates only the prediction slot.

A linear-attention block computes $\langle z_i^{\mathrm{tr}}, z\rangle$ via $QK^{\top}$ with suitable projections, uses the value projection to carry the support residuals $-\eta e_i^{(t)}$, and aggregates them according to the query–support inner products. The resulting update written to the query prediction slot realizes

$$\hat{y}^{(t)}(z) \mapsto \hat{y}^{(t+1)}(z)$$

for all queries in parallel. The block parameters are fixed and do not depend on $t$. Exact parameter constructions follow the standard linear-attention derivation in von Oswald et al. (2023).

D.4.3. STEP 3: STACKING $T$ BLOCKS POTENTIALLY EQUALS $T$ GD STEPS (YANG ET AL., 2024; GATMIRY ET AL., 2024)

Applying Step 2 repeatedly for $T$ blocks yields $\hat{y}^{(T)}(z) = W^{(T)}z$ for every query embedding $z$. This proves Proposition D.3.

**Limitations.** The connection between in-context learning and gradient descent is an active research direction. Our proof is mechanistic: it shows that, under a linear attention construction. In particular, our analysis does not cover standard softmax attention, normalization or general nonlinear heads.

# E. Additional Experiments

## E.1. Broader Comparisons on the Full UCR Archive

To further validate the effectiveness of TIC-FM, we provide broader comparisons beyond the main Mantis and MOMENT baselines. Specifically, we include TiCT (Yeh et al., 2025), classical machine learning classifiers such as XGBoost (Chen & Guestrin, 2016) and Logistic Regression (LaValley, 2008), the neural time series model TimesNet (Wu et al., 2022), and five additional TSFM backbones paired with RF classifiers, including TimesFM 1.0 (Das et al., 2023), TimesFM 2.0 (Das et al., 2023), ToTo (Cohen et al., 2024), TiREX (Auer et al., 2025b), and Moirai 1.1 (Woo et al., 2024). As shown in Table 3, TIC-FM achieves the best average accuracy on the full UCR archive. Moreover, TIC-FM (Syn.), which does not use UCR training labels for adapter calibration, achieves the second-best performance and outperforms all additional baselines. These results further support the effectiveness of support-conditioned in-context inference for time series classification, while TIC-FM avoids fitting a task-specific classifier at deployment time.

## E.2. Scalability with respect to Supervision Budgets

In this section, we study the question: **How performance changes as more labeled examples become available, and whether TIC-FM remains competitive without any parameter updates.**

*Table 3.* **Broader comparison on the full UCR archive.** We include additional classical classifiers, time series models, and TSFM backbones with RF classifiers. Results are average accuracy over the full UCR archive. The best result is in **bold**, and the second best is underlined.

| Method | Avg Acc |
|---|---|
| Logistic Regression | 66.24% |
| TimesFM 1.0 + RF | 67.65% |
| TimesNet | 68.97% |
| XGBoost | 69.17% |
| TimesFM 2.0 + RF | 70.30% |
| ToTo + RF | 71.57% |
| TiREX + RF | 74.57% |
| Moirai 1.1 (Base) + RF | 77.89% |
| TiCT | 79.17% |
| TIC-FM (Syn.) | 79.75% |
| **TIC-FM** | **80.01%** |

Since none of the models are pretrained on the UCR test split, we build this protocol solely from the official test split. For each fraction $\alpha \in \{10\%, 20\%, ..., 60\%\}$, we use stratified sampling to select $\alpha$ of the test examples as labeled context, and evaluate on the remaining $(1 - \alpha)$. We compare TIC-FM against Mantis and MOMENT using the probing heads from their original papers (RF for Mantis; SVM for MOMENT), and also report swapped heads (Mantis+SVM and MOMENT+RF) to control for classifier choice (further details are provided in Appendix F.4.2).

As shown in Figure 3, TIC-FM attains higher average accuracy across all evaluated label fractions than the baselines. As the proportion of labeled examples increases from very limited supervision to more moderate budgets, the mean accuracy of TIC-FM improves steadily, indicating favorable

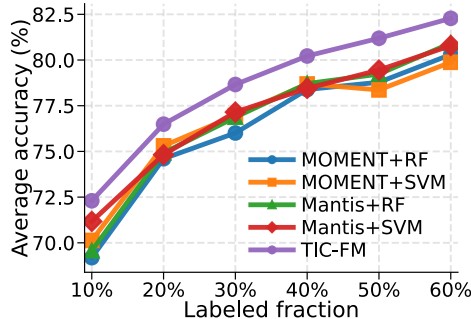

*Figure 3.* **Scalability analysis with labeled data fractions.**

scalability with respect to label availability. In contrast, freeze backbone and classifier baselines do not consistently exhibit monotonic or stable gains as supervision increases. For instance, MOMENT+SVM shows limited improvement and even noticeable fluctuations over certain labeling ranges. These results suggest that the in-context inference paradigm is more robust in leveraging additional labeled information. When labeling is relatively affordable and more labeled context examples can be provided, TIC-FM effectively absorbs the extra supervisory signal and continues to improve predictive performance. When labels are extremely scarce, TIC-FM still benefits from training-free conditioning on the available context set and strong generalization, leading to superior performance. Overall, TIC-FM performs robustly across a broad range of labeling budgets, making it a practical choice under varying annotation constraints.

### E.3. Impact of Context Window Size

We study the question: **How does TIC-FM scale with the context length with fixed query set?**

Unlike Section E.2, which varies the labeled fraction and evaluates on the remaining unlabeled portion, here we isolate the effect of longer contexts and test whether performance saturates. We focus on the three largest multiclass UCR datasets, Crop, ElectricDevices, and ECG5000. We first construct a fixed query set $Q$ by stratified sampling 10% of $\mathcal{D}_{\text{te}}$, ensuring at least one example per class, and keep $Q$ unchanged throughout. The remaining examples constitute a disjoint context pool $P = \mathcal{D}_{\text{tr}} \cup (\mathcal{D}_{\text{te}} \setminus Q)$. Let $C$ denote the number of classes and define the base context size as $N_0 = 10C$, corresponding to roughly ten labeled examples per class on average.

We then vary the context budget as $N_{\text{ctx}} = mN_0$ with $m \in \{1, 5, 10, 15, 20\}$, using class-balanced sampling from $P$ (additional details are provided in Appendix F.2.2).

As illustrated in Figure 4, TIC-FM exhibits a consistent monotonic improvement in accuracy as the context budget $N_{ctx}$ expands from $N_0$ to $20N_0$, with the most pronounced gains occurring in the early low-data regime ($N_0 \rightarrow 5N_0$). This scaling trend suggests that TIC-FM can effectively leverage additional labeled context without requiring parameter updates.

### E.4. Ablation Study

We ablate two key components of TIC-FM while keeping the backbone and evaluation protocol fixed. For **TIC-FM w/o ICL**, we replace the in-context classifier with a conventional RF trained on frozen TIC-FM encoder embeddings. For **TIC-FM w/o ensemble aggregation**, we remove cyclic label permutation and use a single forward pass.

Figure 5 shows that removing the in-context classifier causes the largest drop, confirming that support-conditioned in-context inference is the main source of improvement. Removing ensemble aggregation also reduces accuracy, but less substantially, indicating that label-permutation ensembling mainly improves robustness. Overall, TIC-FM performs best when both in-context reasoning and ensemble aggregation are used.

### E.5. Deployment Efficiency

In addition to accuracy, we evaluate deployment efficiency over all 128 UCR datasets. End-to-end latency is measured as the total runtime of a full evaluation episode divided by the number of query instances. For classifier-fitting baselines, this includes feature extraction, classifier fitting, and prediction; for TIC-FM, it includes support-conditioned in-context inference, with the full model additionally including test-time label-permutation ensembling.

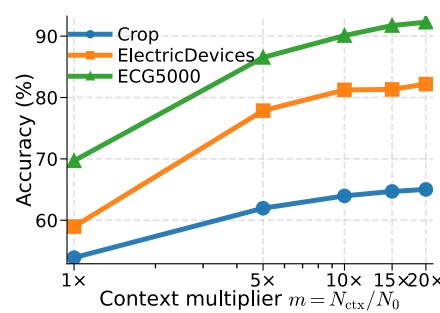

*Figure 4.* **Impact of context length on inference accuracy.**

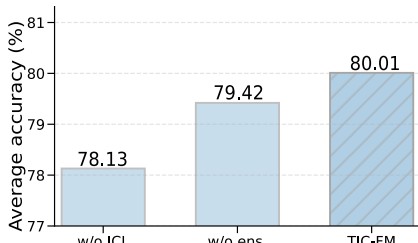

*Figure 5.* **Ablation study on UCR.** In-context inference contributes the largest gain.

As shown in Table 4, TIC-FM uses more peak GPU memory due to support-conditioned reasoning over context and query tokens. However, it benefits from parallel query inference and avoids fitting a task-specific classifier at deployment time. As a result, TIC-FM achieves lower end-to-end latency than both Mantis+RF and MOMENT+SVM. The single-pass variant, TIC-FM w/o ens., is the fastest among all compared methods, showing that the ICL classifier itself is efficient; the full TIC-FM trades additional inference-time ensembling for improved accuracy while still remaining faster than the classifier-fitting baselines.

*Table 4.* **Deployment efficiency on UCR.**

| Method | Mean Acc | ms/query | Mem. (MB) |
|---|---|---|---|
| Mantis + RF | 78.67% | 1.9084 | 1438 |
| MOMENT + SVM | 77.98% | 2.8255 | 760 |
| TIC-FM w/o ens. | 79.42% | **0.4273** | 1907 |
| **TIC-FM** | **80.01%** | 0.9036 | 2488 |

## F. Additional Details and Reproducibility

**Roadmap.** This appendix provides (i) detailed descriptions of baselines (App. F.1); (ii) evaluation protocols for supervision budget scalability and context window size analysis(App. F.2); (iii) implementation and hardware specifications (App. F.3); (IV) additional experimental results(App. F.4).

### F.1. Compared Methods

#### F.1.1. TSFMs

**Mantis.** Mantis is a lightweight foundation model tailored for time-series classification. It adopts an encoder-only architecture that first converts an input sequence into a fixed set of patch-level tokens via a token generator that combines convolutional patching, a differential branch, and patch-wise statistical encoding, and then applies a ViT-style Transformer encoder with a learnable class token to aggregate token information. The model is pretrained in a self-supervised manner

using a contrastive objective, where two stochastic augmentations of the same series form a positive pair and other samples in the batch serve as negatives. In our experiments, we use the 8M-parameter variant of Mantis and initialize it with publicly available pretrained weights.[1]

**MOMENT.** MOMENT is a Transformer-based time-series foundation model pretrained via masked time-series modeling. It partitions a univariate series into non-overlapping patches, embeds them, and randomly masks a subset using a dedicated mask token. A Transformer encoder produces contextualized patch representations, which are fed into a lightweight reconstruction head trained to recover the masked patches under an MSE objective. MOMENT is pretrained on the Time Series Pile, a curated collection covering forecasting, classification, and anomaly-detection datasets (e.g., Informer-style long-horizon benchmarks, the Monash forecasting archive, UCR/UEA classification datasets, and the TSB-UAD anomaly benchmark), using only training splits to mitigate potential contamination. We initialize MOMENT with publicly available pretrained weights.[2]

### F.1.2. CLASSIFIERS ON FROZEN EMBEDDINGS

**Frozen embeddings.** For "a frozen encoder paired with a task-specific classifier" evaluations, we keep the TSFM backbone frozen and extract embeddings for all instances once. Let $z = f(x) \in \mathbb{R}^d$ denote the embedding of a time series $x$, and let $Z_{\text{tr}} = \{(z_i^{\text{tr}}, y_i^{\text{tr}})\}_{i=1}^{N_{\text{tr}}}$ and $Z_{\text{te}} = \{z_j^{\text{te}}\}_{j=1}^{N_{\text{te}}}$ be the resulting train/test embeddings. Unless otherwise stated, we do not apply feature standardization/normalization to the embeddings before fitting any classifier.

**Trained classifiers.** We consider the following lightweight supervised classifiers trained on $Z_{\text{tr}}$: **(i) Random Forest (RF)**, an ensemble of decision trees; **(ii) Linear SVM**, a max-margin linear classifier; and **(iii) one-layer MLP**, implemented as a single linear layer $g_\omega(z) = Wz + b$ trained with cross-entropy.

**Training-free classifiers.** We also report two non-parametric classifiers computed directly in the embedding space: **(i) 1-NN**, which predicts by the nearest training embedding under Euclidean distance, $\hat{y}(z) = y_{i^\star}$ where $i^\star = \arg\min_i \|z - z_i^{\text{tr}}\|_2$; and **(ii) Nearest Centroid (NC)**, which computes class centroids $\mu_c = \frac{1}{|I_c|} \sum_{i \in I_c} z_i^{\text{tr}}$ and predicts $\hat{y}(z) = \arg\min_c \|z - \mu_c\|_2$.

### F.1.3. EVALUATION DETAILS

**Overview.** We evaluate all methods under the freeze backbone and classifier pipeline, as described in the main text, to ensure a fair comparison between trained and training-free classifiers. For feature-based baselines, we obtain pretrained checkpoints of Mantis and MOMENT from the authors' official public repositories. Unless otherwise stated, we freeze all backbone parameters and use each backbone solely as a feature extractor. Given a dataset-specific train/test split, we first compute time-series representations for all samples using the frozen backbone, and then fit a downstream classifier on the training representations only. Final performance is reported on the test split.

**Backbone acquisition and freezing.** For Mantis and MOMENT, we use the authors' released pretrained models and the accompanying preprocessing pipelines provided in their repositories. We do not perform any additional finetuning of these backbones on the evaluation datasets. In all experiments, the backbone is set to evaluation mode, and gradients are disabled, so that the extracted representations are deterministic given the input and random seed.

**Representation extraction.** For each dataset, we transform every time series $x$ into a fixed-dimensional embedding $z \in \mathbb{R}^q$ using the frozen backbone:

$$z = F(x),$$

where $F$ denotes either Mantis or MOMENT. We extract embeddings for both the training split and the test split. To avoid any information leakage, all hyperparameters of the downstream classifier are selected without accessing the test split, and no statistics computed on the test split are used during training (e.g., normalization parameters are computed on the training split and then applied to the test split if needed).

---

[1]https://huggingface.co/paris-noah/Mantis-8M/tree/main
[2]https://huggingface.co/AutonLab/MOMENT-1-base/tree/main

**Downstream classifiers.** On top of the extracted embeddings, we evaluate two categories of classifiers:

- **Trained classifiers.** These methods fit a task-specific classifier on the training embeddings $\{(z_i^{tr}, y_i^{tr})\}$. The classifier is optimized using only the training split of the corresponding dataset. The trained classifier is then applied to test embeddings $\{z_j^{te}\}$ to produce predictions.

- **training-free classifiers.** These methods do not perform gradient-based parameter updates at evaluation time. Instead, they produce predictions for each test instance by conditioning on the labeled training embeddings (e.g., via nearest-neighbor style matching, prompt-based inference, or other non-parametric/ICL-style mechanisms), while keeping all model parameters fixed.

Across both categories, the backbone remains frozen and is never updated on any evaluation dataset.

**Training data usage and leakage prevention.** For every dataset, any fitting/training procedure for downstream classifiers uses only the training split of that dataset. The test split is used exclusively for final evaluation. In particular, when a method conditions on the labeled training set (e.g., training-free prompting/ICL), the labeled set is always the dataset's training split, and no test labels are used at any point.

**Random seeds and reporting.** To account for stochasticity in downstream training (e.g., classifier initialization, minibatch ordering) and any stochastic components in the evaluation pipeline, we run each method on each dataset with **five** different random seeds. We report the **mean test accuracy** across these five runs as the final result for that dataset:

$$\text{Acc} = \frac{1}{5} \sum_{s=1}^{5} \text{Acc}^{(s)}.$$

When applicable, we keep all non-essential factors fixed across seeds (e.g., backbone checkpoint, preprocessing settings) so that the reported variance reflects only the intended sources of randomness.

## F.2. Evaluation Protocols

### F.2.1. SUPERVISION BUDGET SCALING PROTOCOL

**Purpose.** This protocol evaluates how prediction performance scales with the amount of labeled context available at inference time. To avoid any interaction with the official UCR training split in this study, we construct the entire scaling protocol solely from the official UCR test split.

**Pool construction from the official UCR test split.** For each UCR dataset $\mathcal{D}$, we first load the official test set

$$\mathcal{T} = \{(x_i, y_i)\}_{i=1}^{N},$$

and treat it as an unlabeled pool from which we derive both the labeled context set and the query set. For each train-fraction (supervision budget) $\alpha \in \{0.1, 0.2, 0.3, 0.4, 0.5, 0.6\}$, we construct a disjoint split

$$\mathcal{T} = \mathcal{C}_\alpha \cup \mathcal{Q}_\alpha, \qquad \mathcal{C}_\alpha \cap \mathcal{Q}_\alpha = \emptyset,$$

where $\mathcal{C}_\alpha$ is the labeled context set and $\mathcal{Q}_\alpha$ is the evaluation query set.

**Stratified splitting with minimum one per class.** Let $\mathcal{Y}$ be the set of class labels appearing in $\mathcal{T}$, and let $K = |\mathcal{Y}|$. Our splitter enforces (whenever possible) that the context set contains at least one example per class:

$$\forall c \in \mathcal{Y}, \quad |\{(x, y) \in \mathcal{C}_\alpha : y = c\}| \geq 1.$$

Implementation-wise, we perform a two-stage stratified sampling procedure:

1. **Mandatory coverage.** For each class $c \in \mathcal{Y}$, we uniformly sample one index from the class and place it into $\mathcal{C}_\alpha$.

2. **Proportional fill.** If additional context samples are needed, we allocate the remaining budget across classes in proportion to the remaining per-class counts and sample without replacement. Leftover slots (due to rounding) are assigned by largest fractional parts, with a final fallback that assigns remaining slots to any class with available samples.

We also enforce that the query set is non-empty by capping $N_{\text{ctx}} \leq N - 1$. If $N_{\text{ctx}} < K$ under a very small $\alpha$, we set $N_{\text{ctx}} := K$ (still capped by $N - 1$) so that the minimum-one-per-class constraint remains feasible. After sampling, we deterministically shuffle indices within $\mathcal{C}_\alpha$ and $\mathcal{Q}_\alpha$ for bookkeeping.

**Seed control and reproducibility.** We decouple the randomness of data splitting from the randomness of downstream training. Each run is parameterized by: (i) a `split seed` $s_{\text{split}}$ controlling the stratified split of $\mathcal{T}$ into $(\mathcal{C}_\alpha, \mathcal{Q}_\alpha)$, and (ii) five `run seeds` $\{s_r\}_{r=1}^5$ controlling the stochasticity of downstream classifier training. In our implementation, if the run seeds are not explicitly provided, they are generated by drawing five integers from a NumPy RNG initialized with a base seed. Importantly, for a fixed dataset and $\alpha$, we keep the split $(\mathcal{C}_\alpha, \mathcal{Q}_\alpha)$ *fixed across the five run seeds* to isolate variance due to classifier training (e.g., initialization and minibatch order).

**Label re-indexing and class consistency.** For every split, we re-index class labels according to the context labels: we map the unique labels in $\mathcal{C}_\alpha$ to $\{0, \ldots, K-1\}$ and apply the same mapping to $\mathcal{Q}_\alpha$. If any query label is not present in the context label set, the run is invalid; in code this triggers an error. In practice, the minimum-one-per-class constraint prevents such failures except for degenerate cases where some class appears only once in the pool (in which case it must belong to either $\mathcal{C}_\alpha$ or $\mathcal{Q}_\alpha$ under a disjoint split). We log these cases as split warnings.

**Reporting.** For each dataset $\mathcal{D}$ and fraction $\alpha$, we run five trials indexed by run seeds $\{s_r\}_{r=1}^5$ and compute accuracy on the query set $\mathcal{Q}_\alpha$. We report the mean accuracy across the five trials as the final result for $(\mathcal{D}, \alpha)$.

### F.2.2. CONTEXT WINDOW SCALING PROTOCOL

**Testbeds.** Due to the large sample requirements, we use the three largest UCR datasets: Crop, ElectricDevices, and ECG5000. Let $\mathcal{D}_{\text{tr}}$ and $\mathcal{D}_{\text{te}}$ denote the official training and test splits.

**Query set and context pool.** We construct a fixed query set $Q \subset \mathcal{D}_{\text{te}}$ by stratified sampling such that class proportions match those of $\mathcal{D}_{\text{te}}$, ensuring each class appears at least once. Concretely, $Q$ contains $10\%$ of $\mathcal{D}_{\text{te}}$. The context pool is defined as

$$P = \mathcal{D}_{\text{tr}} \cup \left( \mathcal{D}_{\text{te}} \setminus Q \right),$$

so that $Q$ is disjoint from the context used for inference.

**Context construction (class-balanced sampling).** Given a context budget $N_{\text{ctx}}$, we sample a context set $S_{N_{\text{ctx}}} \subset P$ in a class-balanced manner. Let $\mathcal{Y} = \{1, \ldots, C\}$ and $P_c = \{(x, y) \in P : y = c\}$. We allocate $n_c = \lfloor N_{\text{ctx}}/C \rfloor$ examples per class and distribute the remainder $r = N_{\text{ctx}} - C n_c$ by adding one extra example to $r$ randomly chosen classes. If $|P_c| < n_c$, we use all available samples in $P_c$ and reallocate the remaining budget to other classes while preserving balance as much as possible.

**Budgets and reporting.** We sweep $N_{\text{ctx}} \in \{N_0, 5N_0, 10N_0, 15N_0, 20N_0\}$, where $N_0 = 10C$ (roughly ten labeled examples per class at the smallest budget). For each budget, we repeat context sampling five times with different random seeds and report the average accuracy on the fixed query set $Q$.

### F.3. Implementation and Hardware Specifications

**Software Environment and Reproducibility.** To facilitate reproducibility, the complete source code is publicly available.[3] Our method is implemented using `PyTorch` version 2.9.0 and `Python` 3.10. For the comparative time series foundation models, Mantis and MOMENT, we utilized their respective official open-source implementations and corresponding dependency configurations. The implementation of classifiers (e.g., SVM, RF, KNN) and evaluation metrics relies on `scikit-learn` version 1.7.0. Furthermore, the codebase features a modular design to facilitate the extension of new encoders or in-context adapters.

---

[3] https://anonymous.4open.science/r/TIC-FM-AED4

**Hardware Infrastructure.** We utilized distinct hardware environments tailored to the computational demands of different experimental stages. The pretraining phase was executed on a high-performance computing cluster equipped with $8 \times$ AMD Instinct MI200 GPUs, each providing 64GB memory. In contrast, all subsequent experiments, including both the inference of our proposed method and the evaluation of all comparative baselines, were conducted on a computing node featuring $2 \times$ NVIDIA GeForce RTX 3090 GPUs, each with 24GB of memory.

**Architecture and hyperparameters.** TIC-FM consists of a time series encoder $F_\psi$, a projection adapter $g_\phi$, and an in-context classifier $G_\theta$ with a latent-memory module. The encoder $F_\psi$ takes a univariate input sequence of length 512 and produces a 512-dimensional representation. It first applies a token generator built from two one-dimensional convolution layers with a kernel size of 17, and then aggregates the resulting tokens using a ViT backbone with 6 Transformer layers and dropout set to 0.1. The projection adapter is a 2-layer MLP with LayerNorm, hidden size 1024, GELU, and dropout 0.118 that preserves the feature dimension ($512 \rightarrow 1024 \rightarrow 512$). The in-context classifier uses a 12-block Transformer (4 heads, pre-norm) with a Perceiver-style latent memory of 32 latents and 2 write / 2 read cross-attention blocks; labels are embedded via a one-hot linear map and decoded by an MLP ($512 \rightarrow 1024 \rightarrow C_{\max}$) with $C_{\max} = 10$. Detailed hyperparameters are provided in Table 5.

*Table 5.* **Architecture and key hyperparameters of TIC-FM.**

| Module | Hyperparameter | Value |
|---|---|---|
| Encoder $F_\psi$ | Input length ($L$) | 512 |
| | Feature dim ($d$) | 512 |
| | Token-generator conv layers | 2 |
| | Conv kernel size | 17 |
| | ViT Transformer layers | 6 |
| | Encoder dropout | 0.1 |
| Adapter $g_\phi$ | MLP dims | $512 \rightarrow 1024 \rightarrow 512$ |
| | Nonlinearity | GELU |
| | LayerNorm | yes |
| | Dropout | 0.118 |
| ICL classifier $G_\theta$ | Transformer blocks | 12 |
| | Attention heads | 4 |
| | Norm-first (pre-norm) | yes |
| Latent memory | # latents | 32 |
| | Write layers / Read layers | 2 / 2 |
| Label space | $C_{\max}$ | 10 |

## F.4. Additional Results

### F.4.1. ADDITIONAL COMPARISONS ON THE FULL UCR ARCHIVE

Table 6 reports per-dataset classification accuracy on all UCR datasets for each backbone–classifier combination and TIC-FM. We include these results to complement the aggregate metrics in the main paper (average accuracy and mean rank) and to enable a fine-grained inspection of where improvements arise. In particular, per-dataset reporting helps assess whether gains are broadly distributed across datasets or concentrated in a small subset, and it makes explicit the sensitivity of freeze backbone and classifier evaluation to the choice of classifier.

Each entry is reported as the average accuracy computed over five independent runs with distinct random seeds. This protocol ensures that the reported performance is robust to the stochasticity inherent in classifier initialization and training. For deterministic classifiers under our protocol, the range can be zero. We emphasize that the range is included to expose the variability introduced by classifier training and data-dependent optimization, which is especially pronounced for non-convex heads such as MLP.

Two patterns are noteworthy. First, for a fixed backbone, the relative ordering of classifier heads can change substantially across datasets, indicating that downstream performance is often dominated by the classifier's optimization behavior rather than solely by the frozen representations. Second, TIC-FM attains competitive or best performance on a large fraction of datasets without training a task-specific classifier, supporting our claim that inference-time conditioning provides a more

reliable evaluation pipeline when comparing time series foundation models.

*Table 6.* Per-dataset classification accuracy (average) on the 128 UCR datasets. The best results are in **bold**.

| Dataset | MOMENT | | | | | Mantis | | | | | Ours |
|---|---|---|---|---|---|---|---|---|---|---|---|
| | RF | SVM | MLP | kNN | NC | RF | SVM | MLP | kNN | NC | TIC-FM |
| ACSF1 | **0.8040** | 0.6800 | 0.2560 | 0.7000 | 0.5800 | 0.7820 | 0.4760 | 0.3260 | 0.6800 | 0.5600 | 0.6700 |
| Adiac | **0.7918** | 0.2916 | 0.0261 | 0.7545 | 0.7212 | 0.7253 | 0.7673 | 0.4179 | 0.6547 | 0.5857 | 0.6573 |
| AllGestureWiimoteX | 0.6683 | 0.7129 | 0.3286 | 0.7043 | 0.5457 | 0.6609 | 0.6614 | 0.4060 | **0.7157** | 0.4286 | 0.7000 |
| AllGestureWiimoteY | 0.7023 | **0.7443** | 0.3454 | 0.7429 | 0.5543 | 0.6483 | 0.7186 | 0.3929 | 0.7357 | 0.3686 | 0.7243 |
| AllGestureWiimoteZ | 0.5840 | 0.6157 | 0.2389 | 0.5871 | 0.4171 | 0.6649 | 0.6686 | 0.3863 | 0.6386 | 0.3957 | **0.6686** |
| ArrowHead | **0.8229** | 0.6971 | 0.5577 | 0.7943 | 0.4571 | 0.7166 | 0.8103 | 0.5703 | 0.7543 | 0.5714 | 0.7771 |
| BME | 0.9720 | 0.9800 | 0.6240 | 0.9467 | 0.7933 | 0.9347 | **0.9933** | 0.6253 | 0.9533 | 0.6400 | 0.9533 |
| Beef | 0.7200 | 0.6000 | 0.3867 | 0.5667 | 0.5000 | 0.6533 | **0.7533** | 0.4733 | 0.6333 | 0.4333 | 0.7333 |
| BeetleFly | 0.9300 | **0.9500** | 0.8100 | **0.9500** | **0.9500** | 0.8300 | 0.8500 | 0.6900 | 0.8000 | 0.8000 | 0.9000 |
| BirdChicken | 0.8700 | 0.9000 | 0.7300 | 0.8500 | 0.8000 | 0.9900 | 0.9500 | 0.8800 | 0.9000 | **1.0000** | 0.7500 |
| CBF | 0.9240 | 0.9767 | 0.3313 | 0.9189 | 0.9167 | 0.9889 | **1.0000** | 0.8631 | **1.0000** | 0.9733 | 0.9989 |
| Car | 0.7600 | 0.7500 | 0.2133 | 0.8333 | 0.6333 | 0.7867 | **0.8700** | 0.5133 | 0.8333 | 0.6833 | 0.8333 |
| Chinatown | 0.9429 | 0.9650 | 0.6700 | 0.9329 | 0.9067 | 0.8426 | 0.9096 | 0.6974 | 0.7901 | 0.8630 | **0.9738** |
| ChlorineConcentration | 0.6822 | 0.5716 | 0.5326 | 0.6414 | 0.3104 | 0.6765 | **0.6888** | 0.5380 | 0.6242 | 0.3250 | 0.5656 |
| CinCECGTorso | 0.6662 | **0.7565** | 0.2790 | 0.7014 | 0.5529 | 0.6584 | 0.7539 | 0.4393 | 0.7246 | 0.5768 | 0.6964 |
| Coffee | 0.9429 | 0.8929 | 0.6500 | 0.9643 | 0.8929 | 0.9571 | **1.0000** | 0.7786 | 0.9286 | 0.9286 | **1.0000** |
| Computers | 0.7016 | 0.7280 | 0.6088 | 0.6800 | 0.5360 | 0.7288 | 0.7032 | 0.6560 | 0.6520 | 0.7240 | **0.7480** |
| CricketX | 0.6846 | 0.7154 | 0.1846 | 0.7231 | 0.5564 | 0.7328 | 0.6923 | 0.5667 | 0.7769 | 0.6462 | **0.8103** |
| CricketY | 0.6595 | 0.7308 | 0.1974 | 0.7154 | 0.5154 | 0.7374 | 0.7282 | 0.5544 | **0.8051** | 0.6308 | 0.7897 |
| CricketZ | 0.6846 | 0.7179 | 0.1087 | 0.6692 | 0.6026 | 0.7733 | 0.6795 | 0.5656 | **0.8179** | 0.6615 | 0.8077 |
| Crop | 0.6798 | **0.6994** | 0.4318 | 0.6662 | 0.4567 | 0.6689 | 0.6940 | 0.6395 | 0.6403 | 0.5346 | 0.6514 |
| DiatomSizeReduction | 0.8451 | 0.7092 | 0.3007 | **0.9542** | 0.8235 | 0.8575 | 0.8922 | 0.7745 | 0.9248 | 0.8954 | **0.9542** |
| DistalPhalanxOutlineAgeGroup | 0.7281 | 0.7554 | 0.4676 | 0.7050 | 0.7122 | **0.7885** | 0.7050 | 0.7439 | 0.7698 | 0.6978 | 0.7410 |
| DistalPhalanxOutlineCorrect | **0.8167** | 0.7862 | 0.5833 | 0.7355 | 0.4203 | 0.7543 | 0.7210 | 0.7659 | 0.7391 | 0.6775 | 0.7717 |
| DistalPhalanxTW | 0.6691 | 0.6691 | 0.3022 | 0.5827 | 0.6187 | 0.6820 | 0.6475 | 0.6345 | 0.6043 | 0.5396 | **0.6835** |
| DodgerLoopDay | 0.2975 | 0.4000 | 0.1825 | 0.2875 | 0.2500 | 0.4975 | 0.4975 | 0.3475 | 0.3875 | **0.5125** | 0.4750 |
| DodgerLoopGame | 0.7319 | **0.8406** | 0.6000 | 0.6812 | 0.5797 | 0.7246 | 0.7275 | 0.5696 | 0.7029 | 0.6087 | 0.5942 |
| DodgerLoopWeekend | 0.9043 | 0.9565 | 0.8203 | 0.8623 | 0.8696 | 0.9536 | 0.9710 | 0.9261 | 0.9493 | **0.9783** | 0.9420 |
| ECG200 | 0.8240 | **0.8700** | 0.6400 | 0.8100 | 0.7700 | 0.8220 | 0.8600 | 0.7520 | 0.8200 | 0.7700 | 0.8200 |
| ECG5000 | 0.9372 | **0.9447** | 0.8300 | 0.9222 | 0.8571 | 0.9213 | 0.9118 | 0.8929 | 0.9204 | 0.8051 | 0.9376 |
| ECGFiveDays | 0.7233 | 0.9721 | 0.4971 | 0.8908 | 0.5923 | 0.8997 | **0.9823** | 0.6246 | 0.8316 | 0.7085 | 0.9617 |
| EOGHorizontalSignal | 0.5613 | 0.5552 | 0.2989 | 0.5331 | 0.4696 | **0.5917** | 0.4724 | 0.4729 | 0.5773 | 0.5110 | 0.5801 |
| EOGVerticalSignal | 0.4823 | **0.5166** | 0.3155 | 0.4917 | 0.3978 | 0.4575 | 0.4558 | 0.3785 | 0.4779 | 0.3674 | 0.4834 |
| Earthquakes | 0.7439 | 0.7266 | 0.7482 | 0.6547 | 0.5468 | **0.7482** | 0.7050 | 0.7424 | 0.6906 | 0.4964 | 0.7482 |
| ElectricDevices | 0.7172 | **0.7513** | 0.5291 | 0.6832 | 0.5283 | 0.7228 | 0.7034 | 0.7012 | 0.6995 | 0.5818 | 0.7199 |
| EthanolLevel | 0.4356 | 0.3680 | 0.2848 | 0.3400 | 0.3100 | 0.2980 | **0.4940** | 0.2688 | 0.2560 | 0.2740 | 0.3100 |
| FaceAll | 0.7304 | **0.8077** | 0.2244 | 0.7349 | 0.4195 | 0.7807 | 0.8024 | 0.7564 | 0.7811 | 0.7775 | 0.7444 |
| FaceFour | 0.6886 | 0.7955 | 0.3023 | 0.7727 | 0.7955 | 0.9455 | **0.9659** | 0.7364 | 0.9545 | 0.9545 | 0.9205 |
| FacesUCR | 0.7355 | 0.8546 | 0.1374 | 0.7878 | 0.5541 | 0.8245 | 0.8840 | 0.4589 | **0.8932** | 0.8420 | 0.8893 |
| FiftyWords | 0.6422 | **0.7736** | 0.1266 | 0.6527 | 0.6000 | 0.6295 | 0.7516 | 0.4945 | 0.7275 | 0.6901 | 0.6923 |
| Fish | 0.8640 | 0.8629 | 0.1371 | 0.8686 | 0.8057 | 0.9383 | **0.9451** | 0.6274 | 0.9314 | 0.9143 | 0.8743 |
| FordA | 0.9262 | **0.9417** | 0.8577 | 0.8917 | 0.6841 | 0.8565 | 0.9106 | 0.8602 | 0.7939 | 0.7833 | 0.8962 |
| FordB | **0.8064** | 0.7938 | 0.6469 | 0.7481 | 0.5716 | 0.7341 | 0.7667 | 0.7420 | 0.6988 | 0.6407 | 0.7568 |
| FreezerRegularTrain | 0.8736 | 0.9488 | 0.7036 | 0.8449 | 0.7751 | 0.9354 | **0.9888** | 0.7874 | 0.9225 | 0.7572 | 0.9811 |
| FreezerSmallTrain | 0.7876 | 0.8077 | 0.6589 | 0.7688 | 0.7491 | 0.8022 | **0.8743** | 0.7535 | 0.7702 | 0.7453 | 0.8568 |
| Fungi | 0.9946 | **1.0000** | 0.3118 | **1.0000** | **1.0000** | 0.8022 | 0.8935 | 0.3817 | 0.8280 | 0.8280 | 0.7849 |
| GestureMidAirD1 | 0.6200 | 0.6538 | 0.2600 | 0.5846 | 0.5154 | 0.6462 | **0.7015** | 0.2631 | 0.6231 | 0.5538 | 0.6846 |
| GestureMidAirD2 | 0.5785 | 0.5231 | 0.2385 | 0.5308 | 0.4615 | 0.6138 | 0.5538 | 0.2585 | 0.5615 | 0.5615 | **0.6154** |
| GestureMidAirD3 | 0.3400 | 0.3538 | 0.1492 | **0.3692** | 0.2615 | 0.3400 | 0.3477 | 0.1538 | 0.3231 | 0.2615 | **0.3692** |
| GesturePebbleZ1 | 0.8779 | 0.9128 | 0.1930 | 0.8081 | 0.8140 | **0.9279** | 0.9186 | 0.7023 | 0.9070 | 0.8488 | 0.9070 |
| GesturePebbleZ2 | 0.8823 | 0.9114 | 0.3747 | 0.7405 | 0.8608 | 0.9241 | 0.8734 | 0.7266 | 0.8544 | **0.9494** | 0.8228 |
| GunPoint | 0.9840 | **0.9933** | 0.5440 | 0.9600 | 0.7333 | 0.9693 | 0.9867 | 0.8733 | 0.9800 | 0.9133 | **0.9933** |
| GunPointAgeSpan | 0.9766 | 0.9842 | 0.6424 | 0.9652 | 0.6361 | 0.9911 | **0.9949** | 0.8342 | 0.9905 | 0.9652 | 0.9842 |
| GunPointMaleVersusFemale | 0.9804 | 0.9842 | 0.5911 | 0.9810 | 0.9367 | **0.9968** | 0.9937 | 0.7987 | 0.9873 | 0.8703 | 0.9968 |
| GunPointOldVersusYoung | 0.9721 | 0.9492 | 0.5898 | 0.9683 | 0.5524 | 0.9968 | 0.9968 | 0.8267 | **1.0000** | 0.8603 | **1.0000** |
| Ham | 0.6724 | **0.7048** | 0.5657 | 0.5429 | 0.6476 | 0.6743 | 0.6590 | 0.5295 | 0.5524 | 0.6667 | 0.6571 |
| HandOutlines | 0.9097 | 0.9216 | 0.6405 | 0.8568 | 0.6784 | 0.9249 | **0.9324** | 0.8951 | 0.8757 | 0.6595 | 0.8811 |
| Haptics | 0.4929 | **0.4968** | 0.2136 | 0.4481 | 0.4156 | 0.4721 | 0.4481 | 0.3156 | 0.4416 | 0.4610 | 0.4481 |
| Herring | 0.6156 | 0.5938 | 0.5938 | 0.6094 | 0.5469 | 0.6375 | **0.7031** | 0.6250 | 0.5000 | 0.6719 | 0.6250 |
| HouseTwenty | 0.9445 | **0.9580** | 0.9143 | 0.9076 | 0.8908 | 0.9445 | 0.9496 | 0.8891 | 0.9412 | 0.9328 | 0.9496 |
| InlineSkate | 0.3593 | 0.3236 | 0.1622 | 0.4091 | 0.2345 | 0.3535 | 0.3298 | 0.2058 | 0.3782 | 0.2655 | **0.4273** |
| InsectEPGRegularTrain | 0.9510 | **1.0000** | 0.4763 | 0.9719 | 0.8916 | **1.0000** | **1.0000** | 0.9655 | **1.0000** | **1.0000** | **1.0000** |
| InsectEPGSmallTrain | 0.9309 | 0.9558 | 0.4763 | 0.8876 | 0.8594 | **1.0000** | **1.0000** | 0.9727 | **1.0000** | **1.0000** | 0.9799 |
| InsectWingbeatSound | 0.5391 | **0.5980** | 0.2957 | 0.4525 | 0.4152 | 0.5127 | 0.4651 | 0.3080 | 0.4369 | 0.4636 | 0.5162 |
| ItalyPowerDemand | 0.9320 | **0.9504** | 0.5621 | 0.9281 | 0.6453 | 0.9044 | 0.8939 | 0.7100 | 0.9213 | 0.8639 | 0.9174 |
| LargeKitchenAppliances | **0.8533** | 0.8480 | 0.7547 | 0.8320 | 0.7733 | 0.7904 | 0.8320 | 0.7045 | 0.7467 | 0.6533 | 0.7920 |

*Table 6.* Per-dataset classification accuracy (average) on the 128 UCR datasets (continued). The best results are in **bold**.

| Dataset | MOMENT | | | | | Mantis | | | | | Ours |
|---|---|---|---|---|---|---|---|---|---|---|---|
| | RF | SVM | MLP | kNN | NC | RF | SVM | MLP | kNN | NC | TIC-FM |
| Lightning2 | 0.7148 | 0.7541 | 0.5410 | 0.7705 | 0.5738 | 0.8033 | 0.7541 | 0.6131 | 0.8525 | 0.6885 | **0.8525** |
| Lightning7 | 0.6493 | 0.6849 | 0.2603 | 0.6438 | 0.7123 | 0.7534 | **0.7753** | 0.5397 | 0.6712 | 0.7260 | 0.7534 |
| Mallat | 0.8791 | 0.8768 | 0.1250 | 0.8857 | 0.9313 | 0.8829 | **0.9408** | 0.5167 | 0.9168 | 0.9356 | 0.9326 |
| Meat | 0.8867 | 0.8333 | 0.4000 | 0.8667 | 0.8333 | **0.9333** | 0.7633 | 0.4700 | 0.8500 | 0.9000 | 0.9000 |
| MedicalImages | 0.7361 | 0.7618 | 0.5145 | 0.7118 | 0.3039 | 0.6966 | 0.7053 | 0.5668 | 0.7053 | 0.4961 | **0.7829** |
| MelbournePedestrian | 0.8379 | 0.8421 | 0.4863 | 0.8233 | 0.5469 | 0.8999 | 0.9192 | 0.8376 | 0.8680 | 0.7589 | **0.9582** |
| MiddlePhalanxOutlineAgeGroup | 0.5636 | 0.6299 | 0.1883 | 0.5390 | 0.5779 | 0.5870 | 0.5195 | 0.5494 | 0.5000 | 0.5195 | **0.6364** |
| MiddlePhalanxOutlineCorrect | **0.8323** | 0.6529 | 0.5704 | 0.7388 | 0.6289 | 0.8055 | 0.8110 | 0.7024 | 0.7251 | 0.5911 | 0.8144 |
| MiddlePhalanxTW | 0.5623 | **0.5974** | 0.2727 | 0.4740 | 0.4675 | 0.5260 | 0.4481 | 0.5065 | 0.4870 | 0.3506 | 0.5714 |
| MixedShapesRegularTrain | 0.9235 | 0.9460 | 0.7722 | 0.9200 | 0.8186 | 0.9391 | 0.9431 | 0.8695 | **0.9530** | 0.8899 | 0.9443 |
| MixedShapesSmallTrain | 0.8646 | 0.8874 | 0.5742 | 0.8412 | 0.7996 | 0.8884 | 0.8957 | 0.7485 | 0.9105 | 0.8763 | **0.9122** |
| MoteStrain | 0.8909 | 0.8986 | 0.7455 | 0.8642 | 0.8203 | 0.9059 | 0.8818 | 0.8184 | 0.8834 | **0.9265** | 0.9185 |
| NonInvasiveFetalECGThorax1 | 0.8821 | **0.9033** | 0.0729 | 0.8539 | 0.8326 | 0.6159 | 0.8056 | 0.5421 | 0.4926 | 0.4539 | 0.8504 |
| NonInvasiveFetalECGThorax2 | 0.9122 | **0.9191** | 0.0703 | 0.8779 | 0.8656 | 0.6778 | 0.8412 | 0.5805 | 0.5878 | 0.5033 | 0.8718 |
| OSULeaf | 0.8562 | **0.8926** | 0.1818 | 0.8554 | 0.8099 | 0.8636 | 0.8843 | 0.6256 | 0.8802 | 0.8802 | 0.8430 |
| OliveOil | 0.8667 | 0.4000 | 0.4000 | 0.8333 | 0.7667 | **0.9133** | 0.5133 | 0.3933 | 0.8667 | 0.8667 | 0.6333 |
| PLAID | 0.7024 | 0.7523 | 0.2484 | 0.6909 | 0.2793 | 0.8086 | 0.8436 | 0.4737 | **0.8566** | 0.3110 | 0.7635 |
| PhalangesOutlinesCorrect | **0.8368** | 0.7016 | 0.6131 | 0.7716 | 0.6131 | 0.7699 | 0.7949 | 0.7378 | 0.7179 | 0.6131 | 0.7506 |
| Phoneme | 0.2782 | 0.2764 | 0.1113 | 0.2315 | 0.1493 | **0.3270** | 0.2608 | 0.1905 | 0.2716 | 0.2384 | 0.3149 |
| PickupGestureWiimoteZ | 0.6880 | 0.7200 | 0.3280 | 0.6600 | 0.6800 | **0.7920** | 0.7360 | 0.4520 | 0.7200 | 0.6800 | 0.7200 |
| PigAirwayPressure | 0.1135 | 0.0817 | 0.0481 | 0.1106 | 0.0721 | 0.4606 | 0.4548 | 0.1279 | 0.4423 | 0.4808 | **0.7067** |
| PigArtPressure | 0.7923 | 0.8606 | 0.2337 | 0.8510 | 0.7644 | 0.8885 | 0.7558 | 0.2077 | 0.9038 | 0.8365 | **0.9375** |
| PigCVP | 0.7317 | 0.7981 | 0.2654 | 0.7788 | 0.7500 | 0.7644 | 0.7904 | 0.2933 | 0.8173 | 0.7692 | **0.8750** |
| Plane | **1.0000** | **1.0000** | 0.0952 | **1.0000** | **1.0000** | **1.0000** | **1.0000** | 0.9162 | **1.0000** | **1.0000** | **1.0000** |
| PowerCons | 0.8922 | 0.9000 | 0.7067 | 0.8389 | 0.7278 | 0.9144 | 0.9189 | 0.8000 | 0.9333 | 0.8500 | **0.9611** |
| ProximalPhalanxOutlineAgeGroup | 0.8400 | 0.8488 | 0.4878 | 0.8098 | 0.8488 | **0.8576** | 0.8390 | 0.8517 | 0.7415 | 0.8146 | 0.8341 |
| ProximalPhalanxOutlineCorrect | **0.8763** | 0.7595 | 0.6838 | 0.8351 | 0.6392 | 0.8131 | 0.8419 | 0.7766 | 0.7663 | 0.6460 | 0.7801 |
| ProximalPhalanxTW | 0.8078 | **0.8098** | 0.3512 | 0.7415 | 0.7366 | 0.7659 | 0.6829 | 0.7532 | 0.7073 | 0.6098 | 0.7902 |
| RefrigerationDevices | 0.5392 | 0.4987 | 0.5163 | 0.4720 | 0.5013 | 0.5019 | **0.5413** | 0.4923 | 0.4747 | 0.5227 | 0.5387 |
| Rock | 0.6640 | 0.7400 | 0.5400 | 0.7600 | 0.5200 | 0.7640 | 0.8200 | 0.5960 | **0.8400** | 0.7200 | 0.6400 |
| ScreenType | 0.5056 | **0.5360** | 0.4123 | 0.4827 | 0.4240 | 0.4464 | 0.4373 | 0.4133 | 0.3787 | 0.4213 | 0.5173 |
| SemgHandGenderCh2 | 0.7443 | 0.7617 | 0.6317 | 0.6767 | 0.6567 | 0.8937 | 0.8700 | 0.7253 | 0.8850 | 0.7000 | **0.9183** |
| SemgHandMovementCh2 | 0.3876 | 0.4200 | 0.3084 | 0.3511 | 0.3289 | 0.7209 | 0.6644 | 0.4276 | 0.7222 | 0.4378 | **0.7489** |
| SemgHandSubjectCh2 | 0.5956 | 0.6556 | 0.4853 | 0.5311 | 0.5178 | 0.8031 | **0.8578** | 0.6018 | 0.8156 | 0.5156 | 0.8400 |
| ShakeGestureWiimoteZ | 0.9040 | 0.9000 | 0.6160 | 0.8000 | 0.9200 | 0.8840 | 0.8840 | 0.6720 | 0.8800 | 0.8800 | **0.9400** |
| ShapeletSim | 0.9322 | **0.9722** | 0.8511 | 0.9333 | 0.9000 | 0.9200 | 0.9222 | 0.7433 | 0.9056 | 0.9056 | 0.7944 |
| ShapesAll | 0.8610 | **0.8733** | 0.1040 | 0.8600 | 0.7650 | 0.8140 | 0.8317 | 0.7310 | 0.8650 | 0.8050 | 0.8483 |
| SmallKitchenAppliances | 0.7819 | 0.7440 | 0.6395 | 0.6667 | 0.6587 | 0.8112 | 0.7920 | 0.7989 | 0.7467 | 0.7920 | **0.8213** |
| SmoothSubspace | 0.9267 | **0.9667** | 0.6840 | 0.9000 | 0.8667 | 0.9080 | 0.9333 | 0.7173 | 0.8467 | 0.9267 | 0.9333 |
| SonyAIBORobotSurface1 | 0.8735 | 0.8968 | 0.4293 | 0.8985 | **0.9151** | 0.7704 | 0.8270 | 0.6296 | 0.8087 | 0.9018 | 0.7188 |
| SonyAIBORobotSurface2 | 0.9123 | **0.9570** | 0.6170 | 0.9129 | 0.8982 | 0.8306 | 0.9081 | 0.7343 | 0.8898 | 0.8416 | 0.8804 |
| StarLightCurves | 0.9763 | 0.9733 | 0.7364 | 0.9649 | 0.8738 | 0.9761 | 0.9648 | 0.9731 | 0.9703 | 0.9526 | **0.9796** |
| Strawberry | **0.9654** | 0.9216 | 0.6432 | 0.9595 | 0.5568 | 0.9503 | 0.9486 | 0.8654 | 0.9378 | 0.7892 | 0.9405 |
| SwedishLeaf | 0.9158 | 0.9376 | 0.0531 | 0.9072 | 0.8240 | 0.9274 | **0.9424** | 0.8125 | 0.9056 | 0.8880 | 0.9296 |
| Symbols | 0.9622 | 0.9678 | 0.1906 | 0.9608 | 0.9558 | 0.9574 | **0.9869** | 0.8953 | 0.9779 | 0.9628 | 0.9598 |
| SyntheticControl | 0.9340 | 0.9633 | 0.6500 | 0.9233 | 0.8633 | 0.9753 | 0.9867 | 0.9193 | 0.9733 | 0.9733 | **0.9933** |
| ToeSegmentation1 | 0.9351 | 0.9386 | 0.8316 | 0.9035 | 0.8772 | **0.9649** | 0.9605 | 0.8825 | 0.8991 | **0.9649** | 0.9035 |
| ToeSegmentation2 | 0.9077 | 0.9231 | 0.9138 | 0.9385 | 0.9077 | 0.9200 | **0.9462** | 0.8631 | **0.9462** | 0.9385 | 0.9077 |
| Trace | **1.0000** | **1.0000** | 0.1900 | **1.0000** | 0.9900 | **1.0000** | **1.0000** | 0.8640 | **1.0000** | **1.0000** | **1.0000** |
| TwoLeadECG | 0.9861 | 0.9956 | 0.4997 | 0.9789 | 0.9464 | 0.9961 | **0.9963** | 0.7716 | 0.9860 | 0.9877 | 0.9903 |
| TwoPatterns | 0.8866 | 0.9838 | 0.4228 | 0.8293 | 0.7540 | 0.8708 | 0.9670 | 0.8007 | 0.8350 | 0.7275 | **0.9852** |
| UMD | 0.9806 | 0.9792 | 0.6542 | 0.9861 | 0.8194 | 0.9694 | 0.9931 | 0.7472 | 0.9931 | 0.6667 | **0.9931** |
| UWaveGestureLibraryAll | 0.8257 | **0.9227** | 0.4342 | 0.8205 | 0.6309 | 0.8382 | 0.8814 | 0.8012 | 0.8504 | 0.7741 | 0.8889 |
| UWaveGestureLibraryX | 0.7518 | 0.7954 | 0.4577 | 0.7406 | 0.6667 | 0.7614 | 0.7647 | 0.7249 | 0.7507 | 0.7164 | **0.8132** |
| UWaveGestureLibraryY | 0.6901 | 0.7281 | 0.3266 | 0.6616 | 0.5103 | 0.6747 | 0.6647 | 0.6386 | 0.6907 | 0.6287 | **0.7426** |
| UWaveGestureLibraryZ | 0.7061 | 0.7398 | 0.3777 | 0.6904 | 0.5771 | 0.7225 | 0.7365 | 0.6853 | 0.7083 | 0.6672 | **0.7661** |
| Wafer | 0.9789 | 0.9974 | 0.8921 | 0.9825 | 0.8251 | 0.9903 | **0.9976** | 0.9581 | 0.9903 | 0.8214 | 0.9916 |
| Wine | 0.6667 | 0.5000 | 0.5000 | 0.5741 | 0.5185 | 0.7667 | 0.5000 | 0.5000 | 0.5926 | 0.7037 | **0.8704** |
| WordSynonyms | 0.5404 | 0.6160 | 0.2194 | 0.5940 | 0.2633 | 0.5448 | 0.6285 | 0.3273 | **0.6928** | 0.4734 | 0.6176 |
| Worms | **0.8442** | 0.7792 | 0.4286 | 0.7662 | 0.7922 | 0.6260 | 0.6753 | 0.5039 | 0.6104 | 0.7143 | 0.6753 |
| WormsTwoClass | **0.8364** | 0.8052 | 0.5714 | 0.8312 | 0.7532 | 0.7922 | 0.7818 | 0.6753 | 0.6883 | 0.6234 | 0.7662 |
| Yoga | 0.8057 | 0.7300 | 0.5357 | 0.8400 | 0.5573 | 0.8099 | 0.7850 | 0.6311 | 0.8267 | 0.6167 | **0.8527** |

### F.4.2. EXTENDED ANALYSIS OF SCALABILITY WITH SUPERVISION BUDGETS

In Figure 6, we present the training-fraction scaling curves for the baseline configurations that are omitted from the main text (Figure 3) for clarity. As illustrated, TIC-FM consistently achieves the highest accuracy across all label fractions,

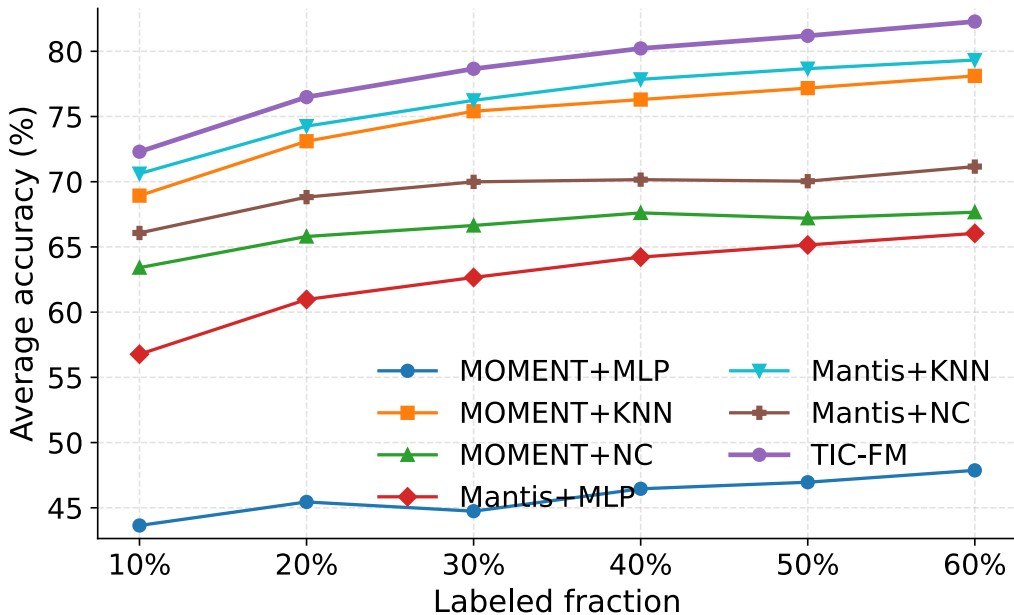

*Figure 6.* **Extended scalability analysis with varying labeled data fractions.** This figure complements Figure 3 by illustrating the performance of TIC-FM against the full set of baseline configurations, including parametric (e.g., MLP) and non-parametric (e.g., KNN, NC) classifiers omitted from the main text. **Observations:** TIC-FM consistently outperforms all baseline variants across all supervision budgets.

demonstrating its robustness compared to both parametric (e.g., MLP) and non-parametric (e.g., KNN, NC) classifiers. Detailed numerical results for all methods under varying supervision budgets are provided in Table 7.

*Table 7.* Average accuracy under different labeled fractions on the UCR test split.The best results are in **bold**.

| Method | 0.1 | 0.2 | 0.3 | 0.4 | 0.5 | 0.6 |
|---|---|---|---|---|---|---|
| MOMENT+RF | 0.6919 | 0.7460 | 0.7601 | 0.7839 | 0.7878 | 0.8030 |
| MOMENT+SVM | 0.7013 | 0.7530 | 0.7696 | 0.7869 | 0.7836 | 0.7988 |
| MOMENT+MLP | 0.4364 | 0.4545 | 0.4474 | 0.4646 | 0.4696 | 0.4787 |
| MOMENT+KNN | 0.6893 | 0.7310 | 0.7540 | 0.7630 | 0.7718 | 0.7811 |
| MOMENT+NC | 0.6341 | 0.6580 | 0.6665 | 0.6761 | 0.6720 | 0.6766 |
| Mantis+RF | 0.6957 | 0.7492 | 0.7687 | 0.7871 | 0.7923 | 0.8092 |
| Mantis+SVM | 0.7117 | 0.7482 | 0.7715 | 0.7845 | 0.7945 | 0.8079 |
| Mantis+MLP | 0.5677 | 0.6097 | 0.6266 | 0.6422 | 0.6515 | 0.6604 |
| Mantis+KNN | 0.7062 | 0.7426 | 0.7624 | 0.7785 | 0.7867 | 0.7933 |
| Mantis+NC | 0.6607 | 0.6882 | 0.6999 | 0.7016 | 0.7004 | 0.7116 |
| **TIC-FM** | **0.7230** | **0.7649** | **0.7866** | **0.8022** | **0.8119** | **0.8228** |

F.4.3. EXTENDED ANALYSIS OF CONTEXT WINDOW SIZE

Table 8 reports the numerical values corresponding to Figure 4, where the context budget is parameterized by the multiplier $m = N_{\text{ctx}}/N_0$ (with $N_0 = 10C$ in our setup). Overall, increasing the number of labeled context examples consistently improves query accuracy across all three datasets (Crop, ElectricDevices, and ECG5000). This finding supports the central premise of TIC-FM, demonstrating that task adaptation is effectively achieved through inference-time conditioning on labeled context rather than by training a task-specific classifier. From an optimization perspective, enlarging the context budget yields a more reliable empirical estimate of the class-conditional structure, thereby enhancing the matching between context and query instances and reducing decision ambiguity.

Across datasets, the gains are most pronounced in the low-context regime. On ElectricDevices, accuracy increases sharply

from 58.97% at $m=1$ to 77.86% at $m=5$, indicating that modest additional supervision can substantially enhance in-context reasoning when labeled support is scarce. Beyond $m=10$, improvements become incremental (81.24% $\rightarrow$ 82.19%), suggesting diminishing returns once the context set becomes sufficiently representative. A similar trend holds for Crop, which improves from 53.91% at $m=1$ to 61.96% at $m=5$, followed by gradual saturation (63.97% at $m=10$ and 65.02% at $m=20$). ECG5000 exhibits the same pattern with a stronger overall baseline, rising from 69.69% at $m=1$ to 86.53% at $m=5$, and continuing to improve more moderately as $m$ increases (90.09% at $m=10$ and 92.27% at $m=20$). This behavior aligns with the analysis in Section **??**: larger context budgets provide more informative conditioning signals that induce optimization-like refinement of activations, with decreasing marginal benefit once contextual evidence becomes strong.

*Table 8.* Context length scaling results (accuracy, %) on Crop and ElectricDevices. The context multiplier is $m = N_{\text{ctx}}/N_0$.

| $m$ | Crop (C=24, $N_0$=240) | ElectricDevices (C=7, $N_0$=70) | ECG5000 (C=5, $N_0$=50) |
|---|---|---|---|
| 1 | 53.91 | 58.97 | 69.69 |
| 5 | 61.96 | 77.86 | 86.53 |
| 10 | 63.97 | 81.24 | 90.09 |
| 15 | 64.70 | 81.32 | 91.73 |
| 20 | 65.02 | 82.19 | 92.27 |

