# OpenReview forum: "Beyond Task-Specific Classifiers: In-Context Inference for Time Series Classification Foundation Models"
_ICML.cc/2026/Workshop/FMSD — FMSD @ ICML 2026 Poster_

### Official Review · Reviewer_qFA1 · 2026-05-17
**Time series classification in a single forward pass via in-context learning**

**Rating:** 7
**Confidence:** 5

**Review:**

## Summary
Time series classification (TSC) is commonly tackled via a pipeline where: (1) representations are extracted from a pretrained TSFM; (2) a downstream classifier is fitted on the target dataset. While effective, the paper argues that this pipeline involves several manual steps to get an effective classifier (i.e. selecting the representation model, the classifier, and hyperparameter tuning). To address this, the paper proposes TIC-FM, a foundation model that classifies query time series in a single forward pass without any training or fine-tuning on the target dataset. This is achieved via in-context learning, where labeled time series serve as support examples and the time series to be predicted is treated as a query. The model is trained in three stages: (1) the time series encoder is pretrained contrastively on synthetic time series generated by CauKer; (2) the in-context classifier is pretrained on synthetic data drawn from a structural causal model prior; (3) both are frozen and a lightweight projection adapter is fine-tuned for 5 epochs on UCR training split. On the UCR archive, TIC-FM achieves the best average accuracy among all compared methods, with particularly strong gains in low-label regimes.

## Strengths
- well motivated: time series classification in a single forward pass is a very interesting paradigm
- good-writing: good structural flow of the writing
- strong empirical results on UCR archive (against the compared baselines though)

## Areas for Improvement
- there are parallel work (https://arxiv.org/abs/2511.19694, https://openreview.net/forum?id=heYrOmVFtY) that proposes solving time-series classification using in-context learning. A comparison would be very beneficial.
- a more complete view of TSC: this paper only compares against TSFMs (Moment and Mantis). There exists also very strong TSC methods and models that spans among classical machine learning, feature engineering, and deep learning. Some examples are: (i) classical ML/feature engineering: Rocket/MiniRocket, Hydra, Catch22; (ii) deep learning: TS2Vec, InceptionTime. Including DTW is also very helpful.
- TSFMs comparison: https://arxiv.org/abs/2506.03128 revealed that pretrained TSFMs for forecasting can be adapted for TSC and showed very competitve results.
- evaluation could be stronger: std-dev/confidence interval of the results should be included (especially that the paper mentioned the experiments are conducted over 5 seeds). Different perspective of the results can be included too (e.g. critical difference diagram, win-rate)
- the evaluation of low-shot settings: the paper compares accuracy at 10% and 15%. However, the number of support samples vary drastically across datasets (e.g. 10% could mean 100 samples for dataset A, but 5 samples for dataset B). Therefore, this results are less solid.
- missing details: how many ensemble members?

## Detailed Comment:
- there seems to be some ablations about the model and inference optimization technique. They can be very insightful. Given that this is a workshop paper, I suggest saving some spaces in the method section and mention the key findings. I like the method, but I would also want to know what components are important.

## Justification of Score
- the method introduced by the paper is interesting.
- promising empirical results
- good writing (although could use better prioritization of content)
- literature and comparison of baselines can be more comprehensive.

---

### Official Review · Reviewer_Ysd5 · 2026-05-19
**Timely and well-written paper**

**Rating:** 8
**Confidence:** 3

**Review:**

# Summary

The authors propose a method for solving time series classification tasks through in-context learning: they train a foundation model on many real and synthetic datasets to predict the test labels given the entire labeled training set and unlabeled test set as input, and show that this generalizes to unseen datasets. They demonstrate superior performance and data efficiency compared to training classifiers on top of frozen foundation models.

# Strengths

- This seems like a useful method which is in line with trends in other domains, e.g. tabular/language data. Empirically it outperforms their baselines and would also be more practical for lots of real-world use cases.
- The experiments and explanations are fairly thorough.

# Areas for improvement

- The method only works for classification problems. Would be interesting to explore ways to generalize to other settings.
- I think the intro slightly undersells the contribution by focusing too much on what's wrong with training classifiers on frozen foundation models. It would be stronger if you link it to the success of similar methods for e.g. language/tabular data, and possibly re-order the paragraphs so you describe your method first and mention the baselines second.
- Some design decisions seem more complicated than necessary and haven't been ablated. It would be good to ablate these and simplify where appropriate.
- Would be good to explain how hyperparameters were chosen for your method and the baselines, e.g. where was the validation set sampled from? Did you look at the datasets used for testing?
- It seems like the appendices could be tightened up a bit, e.g. appendix C seems to repeat a lot of what B already said.

# Detailed comments

- As far as I can tell, the model is trained in multiple stages: first the feature encoder is trained by itself using contrastive learning, then a classifier is trained on top of the frozen feature encoder, then a "projection adapter" which sits in between these two models is trained with the other models frozen. Is this really necessary? Why can't you just train the whole thing end to end (maybe after contrastive pretraining of the feature encoder)?
- I don't see why the projection adapter is helpful. I'm not sure what you mean by "the encoder embedding space is not necessarily aligned with the token space expected by the classifier" -- I think some references and/or experiments are warranted here to explain/demonstrate this. The classifier starts with a pre-norm Perceiver block and is already trained on the raw outputs of the encoder, no? I don't see why there would be "mismatched feature statistics" or why adding another MLP block between the models would fix this.
- Why are you manually extracting the mean, standard deviation, and first-order difference of the patches? AFAIK everything except the standard deviation can be learned by a single conv filter if it turns out to be helpful, and I don't think it's standard to do this in transformer patchifiers.
- The theory is interesting. I haven't checked it, but the results seem plausible. However, it seems to stop at showing that there exists some instantiation of your system which approximates a classifier, or approximates a gradient descent step. It would be more compelling if you could show the system should actually learn this things if it perfectly solves your objective. e.g. with normal cross-entropy classification you should get the Bayes optimal classifier if you have infinite data, infinitely expressive model, perfect optimizer. Can you show that this is also true of your system when you have an infinitely large context set?

# Justification of score

I think this is well-written and timely, and will be interesting to many people at the workshop. I didn't find evidence of methodological errors. While some aspects of the paper could seemingly benefit from more polish, these are somewhat nitpicky and don't undercut the contribution.

---

### Official Review · Reviewer_ytXN · 2026-05-20
**A novel approach to foundation models for time series classification**

**Rating:** 7
**Confidence:** 5

**Review:**

### Summary
The authors present TIC-FM a method for time series classification which moves from the standard “frozen-backbone classifier-fitting” pipeline to single step inference using in-context inference.
### Strengths
The authors present a new way of handling time series classification using in-context learning using a single joined pipeline which does not require to train a task specific classifier for each new setting which seemingly leads to better downstream performance compared to other foundation models.

The evaluation compares to relevant current foundation models in Mantis and MOMENT on different types of classifiers, which additionally provides interesting insights about the performance of different classification heads for these foundation models.

The performance on UCR is generally strong compared to the two baselines. Further, it is very strong to see that the performance of the purely synthetic model is not far of the model which was still trained on UCR, indicating that eventually SOTA performance of purely synthetic models could be possible.
### Areas for Improvement
A more detailed evaluation, comparing to more methods like TS2Vec, or NuTime which show very competitive performance to MOMENT and Mantis (in the respective papers) could help to better understand the true potential improvement of the approach.

While the authors argue that the conventional multi-step evaluation pipeline is suboptimal due to per-dataset optimization and the entanglement of representation quality with classifier choice, the proposed approach still optimizes feature encoder and classifier separately in the pre-training stage. A detailed evaluation on whether optimizing feature encoder and classifier jointly during pre-training could generate interesting insights whether the multi-stage approach is actually suboptimal or rather if there is any benefit from optimizing both jointly.
### Detailed Comments
As mentioned above, the evaluation could be more extensive to give deeper insights on the benefits of the approach.

The training of the classifier $G_{\theta}$ is unclear, specifically the synthetic pre-training. It is mentioned that this module can be pre-trained purely on SCM data which seems to refer to a tabular prior here. Does the pre-training step for $G_{\phi}$ essentially refer to training a tabular foundation model here, where we then only keep the encoder? If that is the case, it would be very interesting to evaluate whether we can just use encoders of existing TFMs such as TabPFN or TabICL and use them as classifiers in the pipeline.

### Score
I find the paper to produce interesting findings, and fitting for the topic of the workshop. I am happy to give a score of 7 for this work and hope that the comments help the authors improve their work.